



# Constraining an Eddy Energy Dissipation Rate due to Relative Wind Stress for use in Energy Budget-Based Eddy Parameterisations

Thomas Wilder[1,2], Xiaoming Zhai[1], David Munday[3], and Manoj Joshi[1]

[1]School of Environmental Sciences, University of East Anglia, Norwich, UK
[2]National Centre for Atmospheric Science, Department of Meteorology, University of Reading, Reading, UK.
[3]British Antarctic Survey, High Cross, Madingley Road, Cambridge, UK

**Correspondence:** Thomas Wilder (t.m.wilder@reading.ac.uk)

**Abstract.** A geostrophic eddy energy dissipation rate due to the interaction of the large-scale wind field and mesoscale ocean currents, or *relative wind stress*, is derived here for use in eddy energy budget-based eddy parameterisations. We begin this work by analytically deriving a relative wind stress damping term and a linear baroclinic geostrophic eddy energy equation. The time evolution of this analytical eddy energy in response to relative wind stress damping is compared directly with a baroclinic

eddy in a general circulation model for both anticyclones and cyclones. The dissipation of eddy energy is comparable between each model and eddy type, although the nonlinear baroclinic processes in the numerical model cause it to diverge from the analytical model at around day 150. A constrained dissipation rate due to relative wind stress is then proposed using terms from the analytical eddy energy budget. This dissipation rate depends on the potential energy of the eddy thermocline displacement, which also depends on eddy length scale. Using an array of ocean datasets, and computing two forms for the eddy length scale,

a range of values for the dissipation rate are presented. The analytical dissipation rate is compared with a constant dissipation rate ($10^{-7}$ s$^{-1}$) and is shown to vary widely across different ocean regions. Dissipation rates are found to vary from a 1/4 up to 4 times the constant dissipation rate. These dissipation rates are generally enhanced in the Southern Ocean, but smaller in the western boundaries. This proposed dissipation rate offers a tool to parameterise the damping of total eddy energy in coarse resolution global climate models, and may have implications for a wide range of climate processes.

# 1  Introduction

Satellite altimetry data has revealed an ocean surface scattered with geostrophic eddies (Wunsch and Stammer, 1998). Eddies are highly energetic features, containing 80% of the ocean's kinetic energy, and also exhibit a wide swathe of spatial and temporal scales. They can be found most prominently in the western boundary currents (e.g. Gulf Stream) and Southern Ocean, and are generated primarily via baroclinic instability of the mean flow (Holland and Lin, 1975). In the global ocean,

eddies regulate ocean heat uptake (Zhai and Greatbatch, 2006; Zhang and Vallis, 2013; Griffies et al., 2015), modulate volume transport (Hallberg and Gnanadesikan, 2006; Zhai and Yang, 2022), and influence the exchange of ocean properties between the surface and interior (McGillicuddy et al., 1998; Dove et al., 2022). Faithfully representing eddy dynamics in non-eddy resolving ocean models is therefore integral for accurate future climate projections.





The representation of mesoscale eddies in coarse resolution ocean models is usually carried out using the Gent-McWilliams
(GM) parameterisation (Gent and Mcwilliams, 1990; Gent et al., 1995). The GM scheme advects tracers downgradient, mim-
icking the process of isopycnal flattening by ocean eddies and release of potential energy via baroclinic instability. As a result
of the GM scheme in global ocean models, significant improvements have been made to the ocean circulation (Hirst and Mc-
Dougall, 1996; Gordon et al., 2000). Danabasoglu et al. (1994) implemented the GM scheme in a non-eddy resolving ocean
model and found this produced a sharper thermocline and a reduced Southern Ocean meridional overturning. The scheme used
by Danabasoglu et al. (1994) considered only a constant GM transfer coefficient, $\kappa$, although further studies have devised ana-
lytical and numerically inferred forms of $\kappa$ that depend on space and time (Treguier et al., 1997; Visbeck et al., 1997; Ferreira
et al., 2005). However, the use of these GM transfer coefficients do not produce a realistic energetic flow field. This is because
the potential energy released by GM is lost and not reinjected back into the flow field, and as such ignores classical geostrophic
turbulence theory (Charney, 1971). Indeed, coarse-resolving models that employ the original GM scheme have been found to
display much lower levels of eddy kinetic energy than eddy-resolving models without GM (Kjellsson and Zanna, 2017).

With all this in mind, a new fleet of GM style eddy parameterisations have been developed that aim to be more energetically
consistent (Eden and Greatbatch, 2008; Marshall et al., 2012; Jansen et al., 2019; Bachman, 2019). Eddy energy budget-
based eddy parameterisations define a GM transfer coefficient that varies in space and time through its dependence on total
eddy energy $E$, or, eddy kinetic energy. One such parameterisation is called GEOMETRIC and was developed in Marshall
et al. (2012) and later implemented in ocean circulation models (Mak et al., 2018, 2022b). GEOMETRIC time steps a depth
integrated eddy energy budget to inform the value of a transfer coefficient,

$$\kappa_{gm} = \alpha E \frac{N}{M^2} \tag{1}$$

where $\alpha$ is a tuning parameter, $N$ is the vertical buoyancy frequency, and $M$ is the horizontal buoyancy frequency. The $\kappa_{gm}$
term forms part of the source term for eddy energy since potential energy is released from the mean flow to generate eddies.
Benefits of GEOMETRIC and other eddy energy parameterisations have included the emergence of eddy saturation in the
Southern Ocean (Mak et al., 2017) and the inclusion of a turbulent energy cascade (Jansen et al., 2019), and thus warrant
further investigation.

Whilst energy budget-based eddy parameterisations offer improvements, there are current uncertainties surrounding the
dissipation rate of eddy energy, which will feed back into uncertainties in the GM coefficient. Mak et al. (2022b) looked at
varying dissipation timescales for eddy energy and what impact this has over the global ocean. They found that less damping
of eddy energy led to a reduction in the uptake of heat, whilst the opposite is true for increased damping. The authors attributed
this to a deeper global pycnocline, along with stronger ACC and AMOC transports, ultimately putting the global ocean in a
position to take up more heat. It is therefore necessary to try and constrain an eddy energy dissipation rate to obtain a realistic
projection of the global climate. However, the dissipation of eddy energy is not governed by one single mechanism, but instead
by many different ones (Ferrari and Wunsch, 2009). Examples include, but are not limited to, eddy-wave interaction (Barkan
et al., 2017), bottom drag (Huang and Xu, 2018), and the western boundary graveyard effect (Zhai et al., 2010). This makes the
task of finding a dissipation rate that encompasses all of these processes arduous, although an attempt has been made recently





using a numerical optimisation method (Mak et al., 2022a). We believe tackling this problem from a theoretical stand point could be complimentary to the top-down approach employed by (Mak et al., 2022a).

One important dissipation mechanism of eddy energy is relative wind stress, a process that can directly spin down mesoscale eddies by applying surface friction (Dewar and Flierl, 1987). Relative wind stress is described by

$$\boldsymbol{\tau}_{rel} = \rho_a C_d |\boldsymbol{u}_a - \boldsymbol{u}_0| (\boldsymbol{u}_a - \boldsymbol{u}_0) , \tag{2}$$

where $\rho_a$ is air density, $C_d$ is a drag coefficient that could be a function of wind speed, $\boldsymbol{u}_a$ is the atmospheric wind $10$ m above the ocean surface, and $\boldsymbol{u}_0$ are surface ocean velocities. Relative wind stress is termed so because it uses the relative motion

between wind and ocean current velocities, $\boldsymbol{u}_a - \boldsymbol{u}_0$. In contrast, the absolute wind stress

$$\boldsymbol{\tau}_{rel} = \rho_a C_d |\boldsymbol{u}_a| \boldsymbol{u}_a , \tag{3}$$

neglects the ocean surface current, $\boldsymbol{u}_0$. The inclusion of the ocean surface current in (2) has led to improvements in estimating the wind power input into the large- and small- scale ocean circulation . For example, using relative wind stress has led to a 20-35% reduction in wind power input into the large-scale ocean circulation (Duhaut and Straub, 2006; Hughes and Wilson,

2008), a reduction in equatorial surface current speeds by 30% (Pacanowski, 1987), and damping of eddy kinetic energy by 10-30% (Zhai and Greatbatch, 2007; Munday and Zhai, 2015; Renault et al., 2016b). In addition to these impacts, relative wind stress also influences the global climate system. Wu et al. (2017) looked at the decadal impact of relative wind stress in a global ocean model and found reductions in the Atlantic Meridional Overturning Circulation of around 13% as well as a 0.2 PW decrease in the maximum northward heat transport. Moreover, Renault et al. (2016a) used a regional model to reveal

relative wind stress ability in stabilising the Gulf Stream path, which was found later to be a result of reductions made to the forward and inverse cascade of energy (Renault et al., 2019). It is clear that relative wind stress does have a significant role in the global climate system and as such provides justification for its use in this current work. A further justification comes from availability of ocean observations, meaning that we can utilise these data to infer a global map of the eddy energy dissipation rate.

In this paper we will derive a constrained eddy energy dissipation rate due to relative wind stress damping, validating this approach against a numerical model. In Section 2 we present theory used in this paper and also derive key analytical equations for the dissipation rate. Section 3 provides an overview of the experimental design. Section 4 looks at the evolution of total eddy energy in response to relative wind stress, comparing an analytical and numerical model. The dissipation rate is then presented in Section 5. Section 6 concludes the paper.

**2   Theoretical framework**

**2.1   Deriving an expression for relative wind stress damping**

The first objective of the theoretical framework is to derive an analytical expression that approximates the damping of eddy energy due to relative wind stress. This can be done by making some assumptions on eddy shape and wind profile.





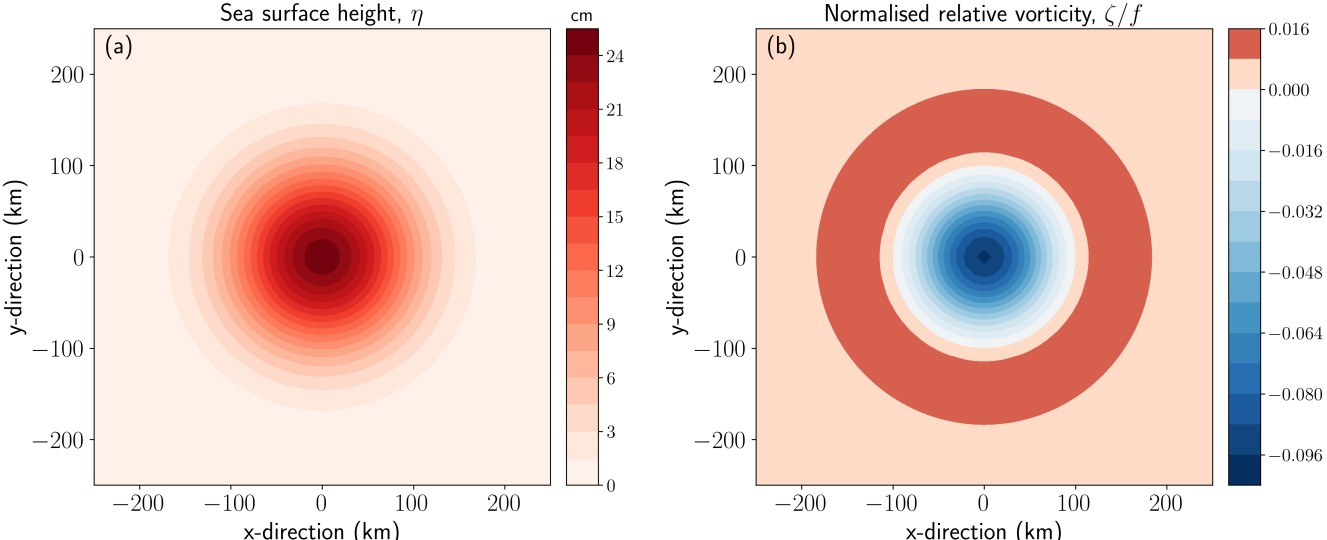

**Figure 1.** Idealised Gaussian eddy with anticyclonic rotation: a) sea surface height (in cm), and b) relative vorticity normalised by Coriolis parameter, $f$. Fields are calculated using parameters: $A = 25$ cm, $R = 100$ km, and $f = 10^{-4}$ s$^{-1}$.

### 2.1.1 An idealised eddy

A comprehensive study by Chelton et al. (2011) revealed mesoscale eddies to have horizontal velocities that are in geostrophic balance

$$\boldsymbol{u}_g = \frac{g}{f}\boldsymbol{k} \times \nabla_h \eta \,, \tag{4}$$

with a sea surface height field that is closely approximated by a Gaussian function

$$\eta(x, y) = A e^{-(x^2 + y^2)/R^2} \,. \tag{5}$$

In (4), $\boldsymbol{u}_g = (u_g, v_g)$ are horizontal geostrophic surface velocities in the zonal and meridional direction, respectively, $g$ is the gravitational constant, $f$ is the Coriolis parameter, $\boldsymbol{k}$ is the vertical unit vector, $\nabla_h$ are horizontal gradients, and $\eta$ is the sea surface height. In (5), $A$ is the eddy amplitude, $x$ and $y$ are zonal and meridional coordinates, and $R$ is the eddy e-folding radius, which is the point of zero relative vorticity. The $\cdot_g$ in (4) implies geostrophic motion. Surface velocities, $\boldsymbol{u}_g$, can then be found by putting (5) in (4), which give analytical velocities in the form

$$(u_g, v_g) = \left( \frac{g}{f} \frac{2Ay}{R^2}, \, \frac{g}{f} \frac{2Ax}{R^2} \right) \eta \,. \tag{6}$$

The eddy described here exhibits a simple circular profile, as shown through sea surface height and relative vorticity in Fig. 1a,b.





### 2.1.2   Relative wind stress

Recall the bulk formula for relative wind stress in (2) given by

$$\boldsymbol{\tau}_{rel} = \rho_a C_d |\boldsymbol{u}_a - \boldsymbol{u}_g|(\boldsymbol{u}_a - \boldsymbol{u}_g) \,,\tag{7}$$

where only the geostrophic velocity component is employed to enable an analytical derivation. The relative wind stress formula in (7) can be simplified by making use of the approximation due to Duhaut and Straub (2006) for the wind stress magnitude

$$|\boldsymbol{u}_a - \boldsymbol{u}_g| \approx |\boldsymbol{u}_a| - \boldsymbol{u}_g \cdot \boldsymbol{i} \,,\tag{8}$$

where $\boldsymbol{i}$ is a unit vector in the direction of the wind. Equation (8) tells us that only the ocean current aligned with the wind
contributes significantly to the wind stress magnitude. A wind profile for $\boldsymbol{u}_a$ is chosen to be uniform in space, blowing zonally west to east, with zero meridional component, i.e. $\boldsymbol{u}_a = (u_a, 0)$. This wind field represents a large-scale atmospheric wind with length scales larger than those of the mesoscale (Duhaut and Straub, 2006). The effect of the eddy current in relative wind stress is presented in Fig. 2a. A dipole pattern of opposing values emerge at each meridional side of the eddy, where the largest values appear near to the eddy radius. The eddy current is able to modify the spatial pattern of wind stress, even with a uniform
background wind field.

### 2.1.3   Wind power input

The next step in deriving the analytical expression for relative wind stress damping is to find the work done by winds on the surface geostrophic motion. This is done by taking the dot product of relative wind stress and surface geostrophic velocities,

$$W_{rel} = \boldsymbol{\tau}_{rel} \cdot \boldsymbol{u}_g \,,\tag{9a}$$

$$W_{rel} = \rho_a C_d |\boldsymbol{u}_a - \boldsymbol{u}_g|(\boldsymbol{u}_a - \boldsymbol{u}_g) \cdot \boldsymbol{u}_g \,,\tag{9b}$$

$$W_{rel} = \rho_a C_d(|u_a|u_a u_g - |u_a|u_g^2 - u_a u_g^2 + u_g^3 - |u_a|v_g^2) \,.\tag{9c}$$

In the above we have made use of Eq. (8). First, we can see the effect of relative wind stress on wind work in Fig. 2b by plotting the difference between relative and absolute wind work ($\boldsymbol{\tau}_{rel} \cdot \boldsymbol{u}_g - \boldsymbol{\tau}_{abs} \cdot \boldsymbol{u}_g$). Interpreting this wind work difference can be achieved by considering the values in Fig. 2a for an anticyclonic eddy (clockwise rotating). The negative wind stress difference
in the north is multiplied by the positive anticyclonic eddy velocity, whilst the positive wind stress difference in the south is multiplied by the negative eddy velocity, and thus the wind work difference is negative everywhere. This shows wind work by relative wind stress is a net sink for a uniform large-scale wind. So we expect an analytical expression for relative wind stress damping to be negative sign definite. We recognise that other wind profiles could exist, though these have not been explored in this current work.
To find the analytical expression, we put analytical equations for geostrophic velocities (6) into Eq. (9c) and integrate over horizontal space in the limits of $x, y \to \pm\infty$

$$P_{rel} = \int\limits_{-\infty}^{\infty} \int\limits_{-\infty}^{\infty} W_{rel} \, dx dy \,,\tag{10}$$





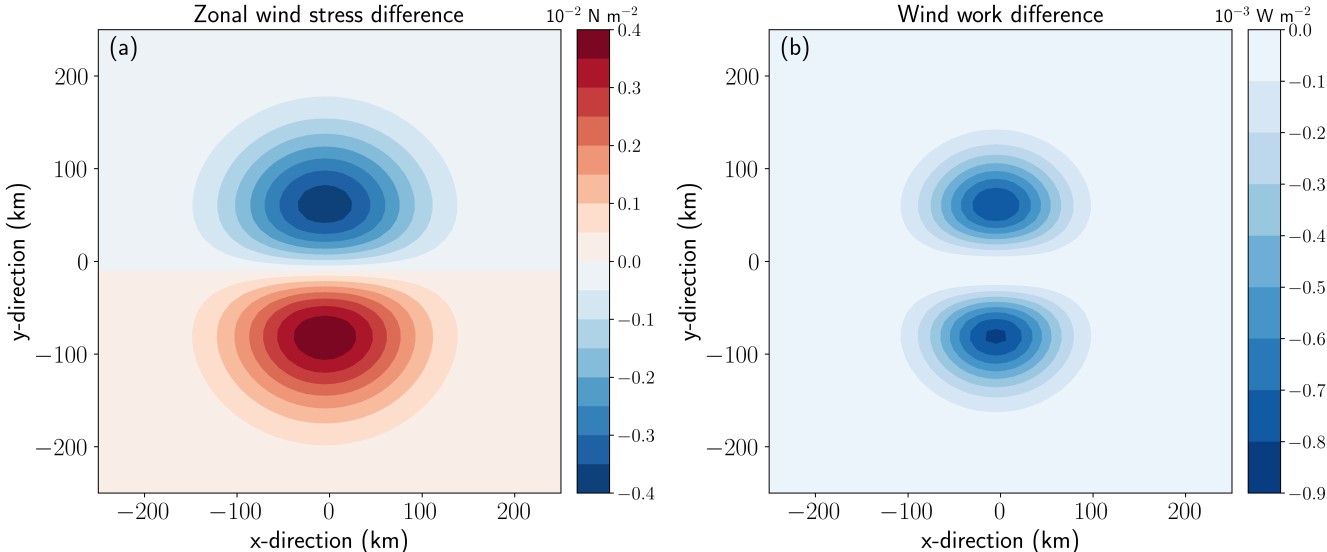

**Figure 2.** Horizontal plan views showing differences between relative and absolute wind stress calculated over an idealised Gaussian anticyclonic eddy: a) difference in zonal wind stress, $\tau_{rel}^{x} - \tau_{abs}^{x}$ (in units $10^{-2}$ N m$^{-2}$), and b) difference in wind work, $W_{rel} - W_{abs}$ (in units $10^{-3}$ W m$^{-2}$). Fields are calculated using parameters: $A = 25$ cm, $R = 100$ km, $f = 10^{-4}$ s$^{-1}$, $u_a = 7$ m s$^{-1}$, $C_d = 1.1 \times 10^{-3}$, and $\rho_a = 1.2$ kg m$^{-3}$.

which gives

$$P_{rel} = -3\rho_a C_d |u_a| \frac{g^2 A^2 \pi}{2f^2} , \qquad (11)$$

where $P_{rel}$ has units kg m$^2$ s$^{-3}$. The analytical equation for relative wind stress damping found here in (11) is analogous to forms suggested by Gaube et al. (2015) and Jullien et al. (2020), although neither carried out a spatial integration. A few things can be inferred from Eq. (11) on $P_{rel}$. First, $P_{rel}$ depends on the magnitude of the wind velocity $u_a$, meaning that damping is independent of the wind direction. Second, $P_{rel}$ is also independent of eddy polarity (sign of $A$) due to its quadratic dependence, implying that anticyclonic or cyclonic eddies will undergo equivalent damping when $A$ is the same in absolute

terms. We also see that $P_{rel}$ does not depend on the eddy e-folding radius, $R$. This is because $R$ cancels out in the integral limits of $\pm\infty$ for this circular eddy, whereby any gains in negative work due to $R$ will be cancelled out by positive work. Finally, with all this in mind, $P_{rel}$ is always negative, informing that relative wind stress will damp eddy energy. If wind power input were to be calculated using absolute wind stress in (3), its spatial integral would equal zero ($P_{abs} = 0$). Overall, this analytical finding is consistent with work by Xu et al. (2016) and Renault et al. (2016b), who find relative wind stress acts as a net sink

of eddy energy.



## 2.2 Describing an analytical eddy

Mesoscale ocean eddies take on a complex vertical structure, making them hard to accurately model. However, studies such as the one by Wunsch (1997) allow us to make reasonable choices in choosing a simple eddy model. Wunsch detailed the variability in eddy kinetic energy (EKE) in the vertical, and found EKE to exist primarily in the barotropic and first baroclinic

modes. These modes can be thought of in terms of their horizontal flow: the barotropic mode has flow that is completely depth-independent; and the first baroclinic mode has flow that is depth-dependent with a zero crossing at depth and zero net flow. Over the global ocean, Wunsch (1997) showed the first baroclinic mode contains the majority of EKE (60-70%), though in some regions, such as south of the Gulf Stream, strong barotropic mode signals were found. Nevertheless, links with the eddy sea surface height and their vertical structure have further been made. It is now widely known that variations in eddy sea

surface height reflect changes in the ocean's thermocline displacement, and thus changes in first baroclinic mode eddy energy (Smith and Vallis, 2001). In this work we proceed with the representation of a singular first baroclinic mode eddy for simplicity.

### 2.2.1 Baroclinic eddy

Two-layer shallow water equations are used to describe the baroclinic eddy

$$\frac{D\boldsymbol{u}_{g1}}{Dt} + f\boldsymbol{k} \times \boldsymbol{u}_{g1} = -g\nabla_h(\eta_1) + \frac{\boldsymbol{\tau}}{\rho_0 h_1} + A_4\nabla_h^4\boldsymbol{u}_{g1} \, , \tag{12a}$$

$$\frac{D\boldsymbol{u}_{g2}}{Dt} + f\boldsymbol{k} \times \boldsymbol{u}_{g2} = -g\nabla_h(\eta_1) - g'\nabla_h(\eta_2) + A_4\nabla_h^4\boldsymbol{u}_{g2} \, , \tag{12b}$$

$$\frac{\partial h_1}{\partial t} + \nabla_h \cdot (h_1\boldsymbol{u}_{g1}) = 0 \, , \tag{12c}$$

$$\frac{\partial h_2}{\partial t} + \nabla_h \cdot (h_2\boldsymbol{u}_{g2}) = 0 \, , \tag{12d}$$

where $\cdot_1$ and $\cdot_2$ is the upper and lower layer, $\eta_2$ is the interface displacement between the two layers, $g' = g(\rho_2 - \rho_1)/\rho_2$ is the reduced gravity - change in acceleration of gravity due to buoyant forces - found using upper and lower layer density,

$h_1 = H_1 + \eta_1 - \eta_2$ and $h_2 = H_2 + \eta_2$ are the respective layer depths of which $H_{1,2}$ is the reference layer depth, and $A_4$ is a constant viscous coefficient that depends on the grid scale and time step. This two-layer model includes the effects of stratification through $g'$, which accounts for the adjustment between the two layers due to the change in density. Equations (12a) and (12b) are momentum equations and Eqs. (12c) and (12d) are continuity equations. The second term on the right hand side of Eq. (12a) is the wind forcing. The third term on the right hand side of Eqs. (12a) and (12b) represents biharmonic

viscosity and is included for completeness as it is present in the numerical model (see Section 3.1).

Before progressing with the derivation of the baroclinic eddy energy equation, some points are discussed first. The two-layer shallow water equations in the form shown in (12) do not immediately describe the baroclinic eddy, rather an ocean with two layers of differing density. It is known that the sea surface height typically reflects the displacement of the main thermocline (Wunsch, 1997). In this case, there exists proportionality between the upper and lower layers in the two-layer analytical model,

and as such the vertical structure of the baroclinic eddy can be described. Following Cushman-Roisin and Beckers (2006), $\eta_1 = \mu\eta_2$ and $\boldsymbol{u}_2 = \lambda\boldsymbol{u}_1$, where $\mu$ and $\lambda$ are proportionality coefficients to be defined, which both provide the dynamical structure



of the eddy through normal modes. Normal modes exhibit wave patterns that depend on these proportionality coefficients, and these are found as follows. Equating together the momentum equations (12b) with (12a) and neglecting dissipation terms gives

$$\lambda = \frac{g\mu + g'}{g\mu} ,$$
(13)

then equating the continuity equations (12d) with (12c) gives

$$\frac{1}{\mu - 1} = \frac{H_2 \lambda}{H_1} .$$
(14)

A quadratic equation for $\lambda$ can be found from (13) and (14)

$$H_2 \lambda^2 + (H_1 - H_2)\lambda - H_1 = 0 .$$
(15)

In (15), there are two solutions for $\lambda$ that relate to the barotropic (BT) and first baroclinic mode (BC1). The BT is described by $\lambda = 1$ and $\mu = H/H_2$, and BC1 is given by $\lambda = -H_1/H_2$ and $\mu = -g'H_2/gH$. A baroclinic eddy is therefore represented by the two-layer model through the use of BC1's $\lambda$ and $\mu$. Whilst $H_1$ is the depth of the upper layer, in BC1 this can also be defined as the first baroclinic mode zero crossing. An example of this mode can be seen in Fig. 1 of Wunsch (1997).

### 2.2.2 Eddy energy equation

The derivation of the two-layer energy equation is done as follows. Equation (12a) is multiplied by $h_1 \boldsymbol{u}_{g1}$, (12b) by $h_2 \boldsymbol{u}_{g2}$, (12c) by $g\eta_1$, and (12d) by $g'\eta_2$, giving the upper and lower layer kinetic and potential energy equations, respectively. The resulting equations are added together to give the total eddy energy equation for an analytical baroclinic eddy

$$\frac{\partial}{\partial t}\Big(\rho_0\big(h_1\frac{1}{2}\boldsymbol{u}_{g1}\cdot\boldsymbol{u}_{g1} + h_2\frac{1}{2}\boldsymbol{u}_{g2}\cdot\boldsymbol{u}_{g2} + \frac{1}{2}g\eta_1^2 + \frac{1}{2}g'\eta_2^2\big)\Big)+$$

$$\nabla_h \cdot \Big(\rho_0(\frac{1}{2}\boldsymbol{u}_{g1}\cdot\boldsymbol{u}_{g1} + g\eta_1)h_1\boldsymbol{u}_{g1} + \rho_0(\frac{1}{2}\boldsymbol{u}_{g2}\cdot\boldsymbol{u}_{g2} + g\eta_1 + g'\eta_2)h_2\boldsymbol{u}_{g2}\Big)$$

$$= \boldsymbol{\tau}\cdot\boldsymbol{u}_{g1} + \rho_0 h_1 \boldsymbol{u}_{g1} A_4 \nabla_h^4 \boldsymbol{u}_{g1} + \rho_0 h_2 \boldsymbol{u}_{g2} A_4 \nabla_h^4 \boldsymbol{u}_{g2} .$$
(16)

In Eq. (16), terms in the top row in order of left to right are: upper layer kinetic energy, lower layer kinetic energy, upper layer potential energy, and lower layer potential energy. Terms in the middle represent the redistribution of kinetic and potential energy by nonlinear advection. In the bottom row: work done by winds on the surface geostrophic motion, upper layer viscous
work, and lower layer viscous work.

We now want to acquire an analytical equation for (16) that we can use to approximate the damping of eddy energy by relative wind stress. To achieve this, Eq. (16) is integrated over space using analytical terms for $\boldsymbol{u}_{g1,2}$ and $\eta_{1,2}$, where the upper layer terms are given in (6) and (5), and the lower layer terms are found using the proportionality coefficients $\lambda$ and $\mu$. First, the combined kinetic and potential energy term from the top row of (16) leads to the following analytical form

$$(KE + PE)_{bc} \equiv E = \rho_0\pi\Big(\big((H_1 - \lambda^2 H_1 + \lambda^2 H)\frac{g^2}{2f^2} + R^2\frac{g}{4} + R^2\frac{g'}{4\mu^2}\big)A^2 + \big((1 - \frac{1}{\mu} + \frac{\lambda^2}{\mu})\frac{2g^2}{9f^2}\big)A^3\Big) ,$$
(17)





which is measured in units of kg m$^2$ s$^{-2}$. Of the two terms that contain $R$, the second one makes up the available potential energy from the lower layer. Since the terms in the middle row of (16) represent the redistribution of energy around the domain, then under no normal flow boundary conditions the integral of this term is zero. Next, the integral of the first term in the bottom row is the wind power input, previously derived in Sect. 2.1. The integral of the two viscous work terms is then given by

$$D_{visc} = -A_4\rho_0 \frac{g^2}{f^2} \left( \pi(1-\lambda^2)\left(H_1 \frac{24A^2}{R^4} + \frac{384A^3}{27\mu R^4}\right) + \lambda^2 H \frac{24A^2}{R^4} - \frac{384A^3}{27R^4}\right) , \tag{18}$$

which is measured in units of kg m$^2$ s$^{-3}$. The viscous dissipation term in (18) is found using the identity

$$\boldsymbol{u}_g A_4 \nabla_h^4 \boldsymbol{u}_g = A_4 \left( \nabla^4 \left(\frac{1}{2}\boldsymbol{u}_g \cdot \boldsymbol{u}_g\right) - \nabla^2(\nabla \boldsymbol{u}_g)^2 - (\nabla^2 \boldsymbol{u}_g)^2 \right) , \tag{19}$$

where the first two terms on the right-hand side represent diffusion, and can be written as the divergence of a flux. These two terms are recognised further as diffusion on kinetic energy and diffusion on gradients of velocity, respectively. Neither of these
terms contribute to viscous dissipation when integrated over the whole domain using no normal flow boundary conditions. The third term contributes to the rate of change of energy through viscous dissipation, and is the term used to form (18).

After integrating Eq. (16) we arrive at an equation in the form

$$\frac{\partial}{\partial t}(KE + PE)_{bc} = P + D_{visc} , \tag{20}$$

where $(KE + PE)_{bc}$ is combined baroclinic kinetic and potential energy per unit volume, $P$ is wind power input, and $D_{visc}$ is
biharmonic viscosity per unit volume. Equation (20) now depends on a few key eddy parameters, in particular eddy amplitude, $A$. For a geostrophic eddy, this means we can take its amplitude and infer the evolution of total eddy energy in response to relative wind stress damping. To do this, the energy equation (20) is integrated forward in time using a fourth-order Runge-Kutta scheme for the first two time steps ($n = 2, 3$), followed by a third-order Adams-Bashforth scheme for time steps $n = 4, \cdots$. Once total eddy energy is found at the next time step $n+1$, eddy amplitude $A$ is recovered from eddy energy $E$ in (17) through
a Newton-Raphson root finder method. The time evolution of analytical eddy energy is then compared with a numerical model in Sect. 4.

## 3 Experimental design

### 3.1 Numerical configuration

The numerical experiments were performed using the hydrostatic MIT general circulation model (Marshall et al., 1997a, b).
Employing this numerical model is done so we can verify whether the analytical wind power input derived in Section 2.1 can sufficiently predict the decay of baroclinic eddy energy due to relative wind stress. The numerical setup was described in detail in Wilder et al. (2022), though we describe some pertinent details along with our attempt to design a continuously stratified model that displays similar characteristics to the analytical two-layer model.

The numerical model is set up on an $f$-plane in a box-like domain spanning 2000 km in each $x$ and $y$ direction with equal
grid spacing of 10 km. The models vertical grid has 91 $z$-levels with spacing of 5 m at the surface and 100 m at depth. The




ocean bottom is flat and a free-slip boundary condition is used, along with no bottom drag. Neglecting bottom drag may have repercussions for the cascade of eddy energy (Scott and Arbic, 2007), however, its neglect means damping by relative wind stress can be isolated in our model. A grid-scale biharmonic viscosity is used for numerical stability purposes as well as to parameterise the dissipation of energy at the smallest of scales.

The baroclinic eddy is initialised using analytical equations. The stratification is given by a 3D temperature field of the form

$$T(x,y,z) = T' e^{-(x^2+y^2)/R^2} e^{-\gamma(z/H_1)} + T_{ref}(z) \,, \tag{21}$$

where $T'$ is the temperature anomaly, $\gamma$ governs the stratification of the water column, $z$ are vertical grid levels, and $H_1$ is the point of zero crossing for horizontal velocities. The background temperature $T_{ref}$ is derived using the linear equation of state
from a reference background density given by

$$\rho_{ref}(z) = \rho_0(1 - N_0^2(z/g)) + 0.5\Delta\rho(1 - \tanh(B(z+H_1)/H)) \,, \tag{22}$$

where $\rho_0$ is a reference density, $N_0$ is a reference buoyancy frequency, $\Delta\rho$ is the difference in density between the surface and bottom, $B$ is the gradient of the density profile, and $H$ is the depth of the ocean. Horizontal velocities are in thermal wind balance

$$\boldsymbol{u}_g(x,y,z) = \frac{g}{f}\boldsymbol{k} \times \left( \nabla\eta + \alpha \int_z^0 \nabla T \, dz \right) \,, \tag{23}$$

where $\boldsymbol{u}_g = (u_g, v_g)$ are zonal and meridional geostrophic velocity components, and $\alpha$ is the thermal expansion coefficient. In (23), the first term in the brackets is surface velocity derived from sea surface height, and the second term is vertical velocity shear derived from thermal wind balance.

So that an adequate comparison of the two-layer baroclinic eddy in Section 2.2.1 can be made with the stratified model
described here, a few parameters in (21) and (22) need to be tuned appropriately. In the two-layer model, the first baroclinic mode has zero net flow in the horizontal, that is flow in the upper layer is countered by an opposing lower layer flow, meaning

$$\int_{-H}^0 \boldsymbol{u}_g \, dz = 0 \,. \tag{24}$$

We minimise net flow in the stratified model by tuning parameters $A$, $T'$, $\gamma$, and $B$. The aim is to achieve a minimal net flow and also have similar eddy properties between each setup, e.g. layer depths and sea surface height. We find the horizontal net
flow in the MITgcm is close to, but not zero. This implies the presence of a barotropic mode component in this setup, which is not too dissimilar to the real ocean (Wunsch, 1997; Arbic and Flierl, 2004). Some key model parameters are shown in Table 1.

When the model is first initialised it is allowed to run for 10 days with zero wind forcing. This allows any inertial waves to die down, and also let the equations of motion form a balance that could be slightly different to geostrophy. After this adjustment phase, the wind forcing is turned on and the model is run for 400 days.





**Table 1.** Key experimental parameters

| Symbol | Value | Description |
| --- | --- | --- |
| $H$ | 4000 m | Ocean depth |
| $\Delta x, y$ | 10 km | Horizontal grid resolution |
| $f$ | $9.3461 \times 10^{-5}$ s$^{-1}$ | Coriolis frequency |
| $A$ | 25 cm | Eddy amplitude |
| $R$ | 100 km | Eddy e-folding radius |
| $\boldsymbol{u}_a$ | 7 m s$^{-1}$ | Wind speed |
| $C_d$ | $1.1 \times 10^{-3}$ | Drag coefficient |
| $\rho_a$ | 1.2 kg m$^{-3}$ | Air density |
| $\rho_0$ | 1026 kg m$^{-3}$ | Reference ocean density |
| $T'$ | 2.5 °C | Temperature anomaly |
| $\gamma, B$ | 1, 3 | Stratification parameters |
| $\Delta\rho$ | 3 kg m$^{-3}$ | Density difference |
| $N_0$ | $10^{-5}$ s$^{-1}$ | Reference buoyancy frequency |
| $H_1$ | 800 m | Upper layer/BC1 zero crossing depth |
| $\rho_2$ | 1026.9 kg m$^{-3}$ | Analytical lower layer density |

## 3.2 Diagnosing model energetics

To validate the evolution of baroclinic eddy energy in the analytical model (Sec. 2.2.1), time-mean quantities of kinetic and potential energy, and wind damping for the continuously stratified MITgcm model need to be defined. The following are mean potential energy, mean kinetic energy, and wind power input

$$PE = -\int_V \frac{g}{2n_0(z)} \overline{\rho^*(x,y,z,t)}^2 \, dV , \quad \text{and} \tag{25}$$

$$KE = \int_V \frac{\rho_0}{2} \left( \overline{u}_g^2 + \overline{v}_g^2 \right) dV , \tag{26}$$

$$P = \int_S \overline{\boldsymbol{\tau}} \cdot \overline{\boldsymbol{u}}_g \, dS , \tag{27}$$

where $^-$ represents a 16 day time-mean, $\rho^*(x,y,z,t) = \rho(x,y,z,t) - \rho_{ref}(z)$ is a density anomaly relative to a constant-in-time reference background density state, $n_0(z)$ is the vertical gradient of $\rho_{ref}(z)$, $u_g$ and $v_g$ are geostrophic velocity components in the zonal and meridional direction, and $\int_V$ is a volume integral. The density field $\rho(x,y,z,t)$ is computed from the MITgcm temperature field, and $\rho_{ref}(z)$ is given in Eq. (22). The use of potential energy anomaly informs how much potential energy can be converted into kinetic energy, as opposed to how much potential energy exists within the stratification. Choosing the potential energy definition in (25) implies a quasi-geostrophic framework and has been used in past studies (von Storch et al., 2012; Chen et al., 2014; Youngs et al., 2017).





### 3.3 Setting up the analytical model

The time evolution of analytical eddy energy is achieved by time-stepping Eq. (20) forward in time. To begin the time-stepping of the analytical model, initial eddy energy and dissipation is found by using data from the MITgcm model run, such as eddy amplitude. Equivalent eddy energy is desired to visualise the rate of decay imposed by relative wind stress. Because the MITgcm setup has been chosen to display similar characteristics to the analytical model, the energetics are thus fixed. To make eddy energy in the analytical model match the MITgcm setup, we modify the analytical lower layer density until

potential energy matches (see Table 1). Kinetic energy is only a small fraction of total eddy energy, so there is less importance in matching this quantity between the analytical and numerical setups. Overall, these details allow us to make a consistent comparison between both setups, and examine more clearly the rate of eddy energy decay by relative wind stress.

### 4 Predicting baroclinic eddy energy decay

In this section we present our first set of results, comparing the time evolution of the analytical and numerical eddy energy

budgets. Figure 3 shows a time-series of domain integrated eddy energy ($KE + PE$) for an anticyclonic (ACE) and cyclonic (CE) eddy. The first thing that can be seen is the initial offset in total eddy energy between the analytical model (Pred) and the numerical model (MIT) in ACE and CE. Here, potential energy is being matched between the analytical and numerical model, and therefore the discrepancy implies that the kinetic energy contribution is not equivalent between Pred and MIT. This kinetic energy mismatch is expected since the two-layer analytical eddy cannot realistically represent the continuously

stratified MITgcm eddy. Focusing on the ACE to begin with, the decay rates of Pred and MIT are fairly consistent (Fig. 3a). In the absolute wind stress case (AW), viscous dissipation erodes the eddy's energy, and so AW Pred and MIT lose 0.1 PJ up to day 150. In the relative wind stress case (RW), there is an additional decay of eddy energy in response to the negative wind power input by relative wind stress (Fig. 5a). Up to day 150, RW Pred loses 0.38 PJ whilst RW MIT loses around 0.4 PJ, relative to day 31 in RW time-series. This damping by relative wind stress is similar because the wind power input in Pred and

MIT is around $-3 \times 10^7$ W.

Beyond day 150, Pred and MIT time-series begin to diverge, with MIT undergoing a sudden reduction in total energy of around 10% over 30 days, whilst Pred continues with a smooth decay. This divergence indicates that MIT is no longer evolving as it initially did, suggesting the eddy is undergoing an instability process and departing from its initial state. In RW, this sudden reduction in total energy also takes place at an earlier timescale. These possible instabilities may also impact the

relative wind power input, since $P_{rel}$ displays a sharp increase in negative wind power input (Fig. 5). An in depth examination of the anticyclonic eddy response can be found in Wilder et al. (2022), and so the finer details are omitted from this discussion. From day 250, the rate of decay in MIT slows for each wind stress and is much more closely aligned with the decay rate in Pred. Inspecting the ACE eddy surface relative vorticity in Fig. 4 illustrates the regime change of the eddy, consistent with the changes seen in the time series (Fig. 3). The ACE under AW and RW is initially coherent at day 125 (Fig. 4a,d), then develops

two outer lobes of stronger cyclonic vorticity by day 200 (Fig. 4b,e), before eventually splitting into two separate anticyclonic





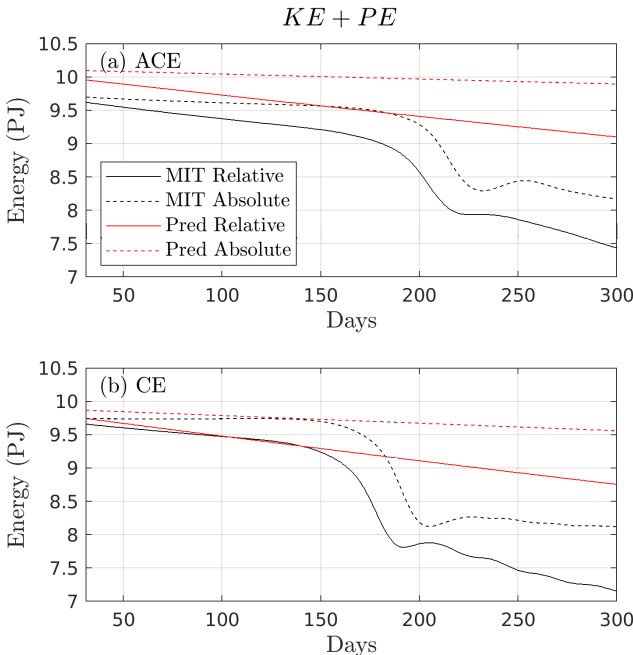

**Figure 3.** Time-series of total eddy energy, $E$ for: a) anticyclone, and b) cyclone. MITgcm shown in black, and predicted shown in red, with absolute wind stress in dashed line and relative wind stress in full line. Units of energy in PJ. MITgcm values are 16 day time-means.

eddies by day 275 (Fig. 4c,f). This process of eddy splitting in baroclinic eddies has been well documented in previous studies (Ikeda, 1981; Dewar et al., 1999), where time-scales vary with parameter values chosen (Mahdinia et al., 2017).

Similar results are also observed for the CE (Fig. 3b). The decay rate in total energy follows roughly the same trajectory as the ACE for 150 days, with more damping taking place in RW Pred and RW MIT due to negative wind power input (Fig.
5b). As discussed earlier, wind power input due to relative wind stress is independent of eddy polarity, so no bias in damping rate should exist. Up to day 130 of the time-series, RW Pred is damped by 0.37 PJ, and RW MIT is damped by 0.26 PJ. The disparity in damping is not a result of unequal dissipation rates by $P_{rel}$ (Fig. 5b), but is a result of energy production in MIT via vertical diffusive processes. Indeed, running a simulation with no eddy, no wind, but with vertical diffusion, did result in potential energy production (not shown). However, why this is more prominent in the cyclonic eddy than anticyclonic has not
been investigated further. After day 150, MIT exhibits a sudden reduction in total energy with each wind stress, which happens earlier in RW. Moreover, in contrast to the ACE, the timescale for this sudden reduction to take place in the CE is around 15-20% shorter. This points to an anticyclone-cyclone asymmetry, which has been recognised in past studies (Chelton et al., 2011; Mkhinini et al., 2014; Mahdinia et al., 2017).

In this section we have compared the evolution of total eddy energy between an analytical and numerical model. The results
tell us that a linearised two-layer analytical model can reasonably explain the evolution of total eddy energy in the MIT simulation. However, the agreement between both models diminishes due to an instability process in the MIT simulation.



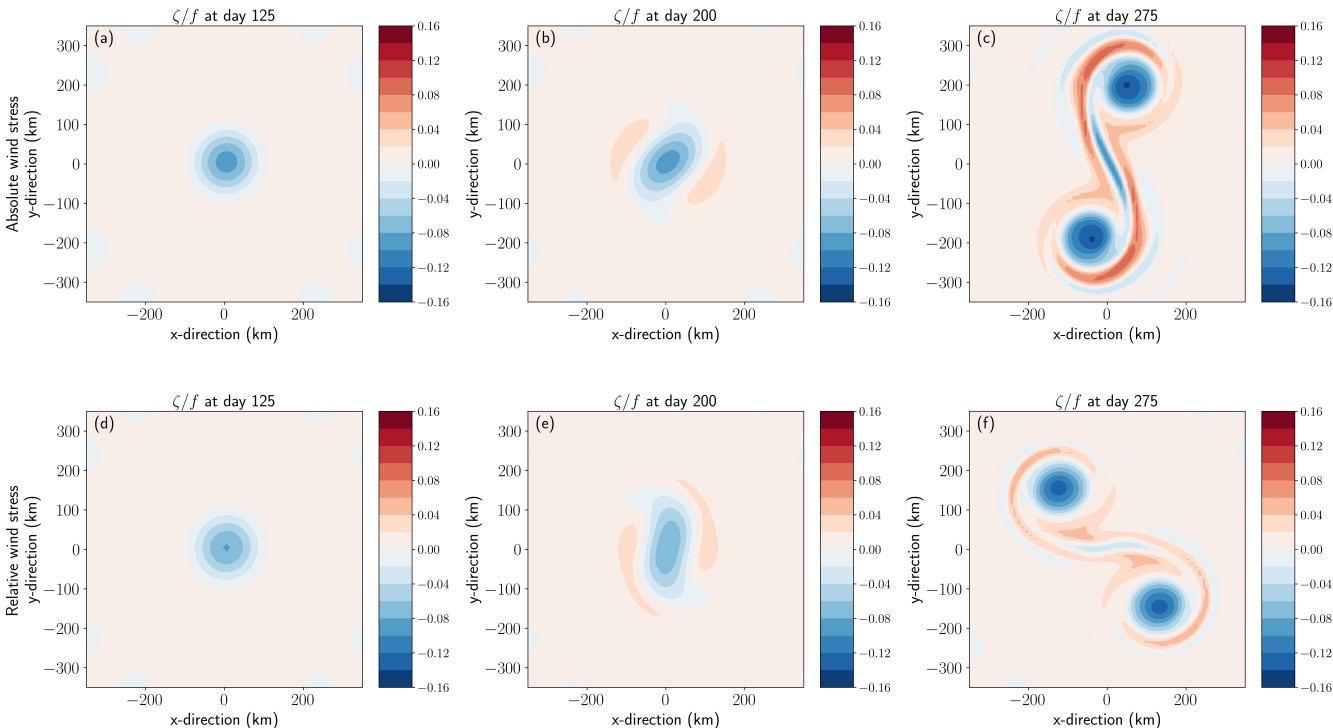

**Figure 4.** Horizontal plan views of MITgcm surface relative vorticity normalised by Coriolis frequency in an anticyclonic eddy for absolute (top) and relative (bottom) wind stress at days: a,d) 125, b,e) 200, and c,f) 275. Fields are calculated using daily mean SSH output from MITgcm simulations.

Nevertheless, we find the time-scale of around $\sim 150$ days for eddy energy agreement to be acceptable, and as such feel confident to propose a constrained eddy energy dissipation rate in Sect. 5.

## 5 A constrained dissipation rate

A dissipation rate due to relative wind stress takes the form

$$\Lambda_{rel} = P_{rel}/E, \tag{28}$$

and can be found by putting the analytical equations for $P_{rel}$ from (11) and $E$ from (17) into the above for a constrained $\Lambda_{rel}$. Since the analytical eddy energy $E$ is made up of several terms, we will simplify $E$. We do this by making a key assumption: that the dominant term that makes up eddy energy is the available potential energy located in the thermocline displacement. Indeed,

potential energy typically outweighs kinetic energy by at least an order of magnitude (von Storch et al., 2012). Therefore, the dissipation rate of eddy energy due to relative wind stress is given by

$$\Lambda_{rel} \approx \frac{6\rho_a C_d |u_a| g^2 \mu^2}{\rho_0 R^2 g' f^2} . \tag{29}$$



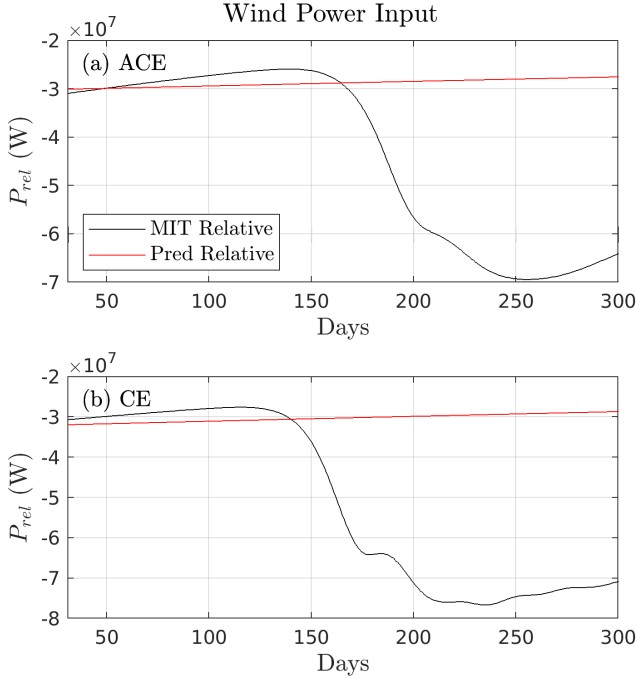

**Figure 5.** Time-series of total wind power input in relative wind stress simulation, $P_{rel}$ for: a) anticyclone and b) cyclone. MITgcm in black line and predicted in red line. Units of power in W. MITgcm values are 16 day time-means.

The dissipation rate $\Lambda_{rel}$ is independent of eddy amplitude due to $P_{rel}$ and $E$ being functions of $A^2$. We see instead that $\Lambda_{rel}$ depends on a few terms that can vary in space, such as wind velocity $u_a$, proportionality coefficient $\mu$, eddy length scale $R$, 340  and reduced gravity $g'$.

## 5.1 Calculating the dissipation rate

We approach the computation of the dissipation rate $\Lambda_{rel}$ by acquiring spatially varying terms, which are $\boldsymbol{u}_a$, $\mu$, $R$, and $g'$. Wind data is now assumed to be the total wind speed $\boldsymbol{u}_a$, rather than just the zonal wind velocity. This is because wind patterns vary in latitude and longitude over the global ocean. Wind speed is taken from the NCEP-NCAR Reanalysis 1 data 345  (Kalnay et al., 1996) and is on a 2 degree grid. The wind speed data is then interpolated onto a 1 degree grid. The remaining terms require temperature and salinity datasets, and these are taken from World Ocean Atlas (Locarnini et al., 2019; Zweng et al., 2019) on a 1 degree grid. Each dataset is made up of long term monthly means over the period 1981 to 2010, which are averaged into seasons June-July-August (JJA) and December-January-February (DJF). The terms $\mu$ and $g'$ are found by solving a Sturm-Liouville eigenvalue problem for the first baroclinic mode using the temperature and salinity fields (see Sect. 350  5.1.1). Arriving at an approximation for the eddy length scale $R$ comes with some uncertainty, and for this reason we establish two forms for $R$. As a result, we will also form two choices for the dissipation rate that will indicate where we might expect the value to fall between. From the eigenvalue problem (Sect. 5.1.1), the first baroclinic Rossby radius of deformation, $R_d$ can be





found, which we take as one choice for $R$. Another choice for $R$ is found by scaling our computed $R_d$ with data from Chelton et al. (2011), Fig. 12, where an e-folding radius $L_e$, and Rossby radius $\hat{R}_d$, are presented over latitude as zonal averages. That
is, our values of $R$ are given as either $R_d$ or $R_d(L_e/\hat{R}_d)$.

### 5.1.1   The eigenvalue problem

Following Xu et al. (2011), the eigenvalue problem takes the form

$$\frac{d}{dz}\left(\frac{f^2}{N^2}\frac{d\phi_n(z)}{dz}\right) + \lambda_n \phi_n(z) = 0, \tag{30}$$

with boundary conditions

$$\frac{d\phi}{dz} = 0 \quad \text{at } z = 0, -H, \tag{31}$$

where $\phi_n(z)$ is the eigenmode, $\lambda_n$ is the eigenvalue, $N(z) = -(g/\rho_0)\partial\rho'(x,y,z)/\partial z$ is the buoyancy frequency, $\rho'(x,y,z)$ is a density anomaly with respect to a reference ocean density, $\rho_0$. Here, $\lambda_n$ is not the same as $\lambda$ defined in Section 2.2.1. The Gibbs SeaWater Oceanographic Toolbox (McDougall and Barker, 2011) is used to calculate $\rho$. The eigenvalue problem (30) is solved using the MATLAB function *dynmodes.m* (Klinck, 2009). From Flierl (1978), the first baroclinic Rossby radius of
deformation is related to the eigenvalue like, $R_d = 1/\sqrt{\lambda_1}$. We subsequently use $R_d$ as one of our choices for the length scale of mesoscale eddies. In addition, we find the zero crossing of the first baroclinic mode, $H_1 = \min|\phi_1(z)|$. Reduced gravity is then defined as $g' = c^2(1 + \phi_1(0)^2)/H_2$, where $c = f/\sqrt{\lambda_1}$ is the first baroclinic gravity-wave phase speed.

### 5.1.2   The contributing terms

Figure 6 displays the terms $\boldsymbol{u}_a$, $\mu$, and $g'$ over the global ocean. Figure 6a,b illustrate the wide variability in space and time
for the wind speed. There is a clear increase in $\boldsymbol{u}_a$ at higher latitudes during each hemispheric winter, whilst a slow down in winds during their summer. The largest wind speeds occur around 90°E in the Southern Ocean, whilst the western boundaries see values a few m s$^{-1}$ slower. In the $\mu$ term, there is a slight variation between seasons, with values the largest in absolute values over the equatorial regions (Fig. 6c,d). The spatial pattern between $g'$ (Fig. 6e,f) and $\mu$ is similar due to $\mu$ depending on $g'$. Across each season, $g'$ remains fairly consistent over the equatorial bands. At higher latitudes, $g'$ varies due to changes in
seasonal stratification.

Figure 7 displays the Rossby radius of deformation ($R_d$) and e-folding scale ($L_e$) used to define the eddy length scale, $R$. Figure 7a,b shows $R_d$, similar to Fig. 6 in Chelton et al. (1998), whereby it decreases in length scale with increasing latitude ($\sim 200$ km to $\sim 10$ km). The e-folding length scale $L_e$ is shown in Fig. 7c,d and similarly varies in latitude, with the largest (smallest) length scales at low (high) latitudes. Comparing $R_d$ and $L_e$, we see that $L_e$ is around 3-4 times bigger than $R_d$
across much of the ocean. Over JJA and DJF periods, there is very little seasonal variability. What isn't clear from the colour saturation is that $L_e$ is smaller than $R_d$ in the equatorial region, by around a factor of a half.





**Figure 6.** Global maps between the latitudes of 70°S and 70°N displaying contributions to the dissipation rate, $\Lambda_{rel}$ for: left) JJA; and right) DJF. In a,b), wind speed, $\boldsymbol{u}_a$ (in units m s$^{-1}$), c,d) proportionality coefficient, $\mu$, and e,f) reduced gravity, $g'$ (in units m s$^{-2}$). The data is caculated from World Ocean Atlas and NOAA datasets over 1981-2010 period.

## 5.2 A global dissipation rate

A global dissipation rate is now presented, culminating from the variable climatology data calculated in Sect. 5.1, along with values from Table 1. Figure 8 shows $\log_{10}(\Lambda_{rel}/10^{-7} \text{ s}^{-1})$ over the global ocean, making it clear where $\Lambda_{rel}$ could be

important for eddy energy dissipation. Each dissipation rate is shown using $R_d$ or $L_e$ for eddy length scale, $R$.





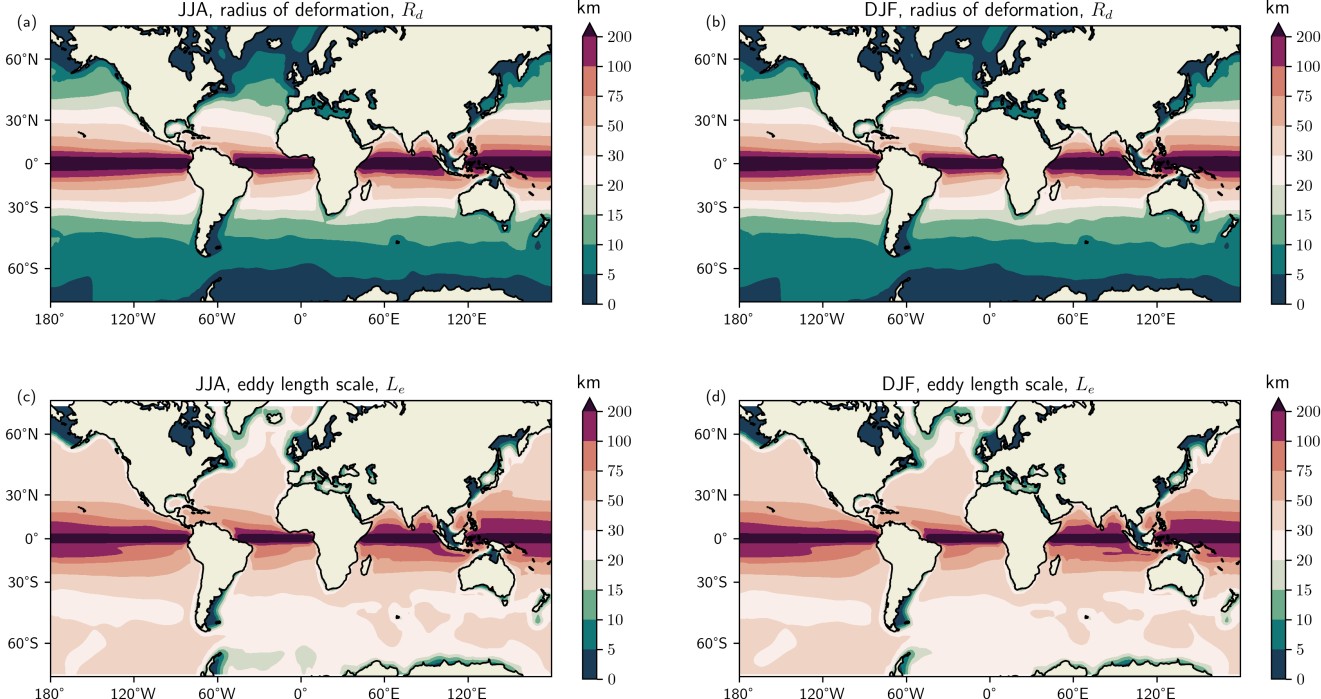

**Figure 7.** Global maps between the latitudes of 70°S and 70°N displaying the eddy length scale (in units km) for: left) JJA; and right) DJF. In a,b), first baroclinic Rossby radius of deformation, $R_d$, c,d) the eddy e-folding length scale, $L_e$. The $R_d$ is calculated from World Ocean Atlas and NOAA datasets over 1981-2010 period, and $L_e$ is computed using data from Chelton et al. (2011). The colorbar has uneven intervals, with spacing increasing with length scale.

Beginning with the Rossby radius of deformation $R_d$, we find $\log_{10}(\Lambda_{rel}/10^{-7} \text{ s}^{-1})$ is largely positive across the global ocean in each season (Fig. 8a,b). In the Southern Ocean we find large values throughout, with $\Lambda_{rel}$ being up to 4 times that of $10^{-7} \text{ s}^{-1}$. This region is known to exhibit important bathymetric features, which impose a control on the Southern Ocean flow (Graham et al., 2012; Munday et al., 2015). For example, the transition from small to large values at 60°W could be in part due to the bathymetry of Drake Passage. We can also see that $R_d$ becomes smaller moving from 120°W to 0°(Fig. 7a,b), contributing to the increase in dissipation rate owing to smaller levels of available potential energy. In the Northwest Atlantic, we see that the zero contour of $\log_{10}(\Lambda_{rel}/10^{-7} \text{ s}^{-1})$ roughly follows the jets separation past Cape Hatteras. Here, the dissipation rate by relative wind stress is similar to the value posed in Mak et al. (2018). From the coast to the basin interior we see that $\Lambda_{rel}$ reduces in size, possibly due to a combination of reductions in $R_d$ and $\boldsymbol{u}_a$, and changes in $g'$, or stratification (Figs. 7a,b and 6). It was shown in Mak et al. (2022b) through their global simulations that the western boundary currents display weaker eddy energy. This is suggested to be because their dissipation rate of $10^{-7} \text{ s}^{-1}$ is too high, and as such the weaker $\Lambda_{rel}$ from the Gulf Stream towards the interior here may hint at that being true. The Kuroshio Extension in the Northwest Pacific also displays values close to zero, but like the Gulf Stream, its values are overall much less pronounced when





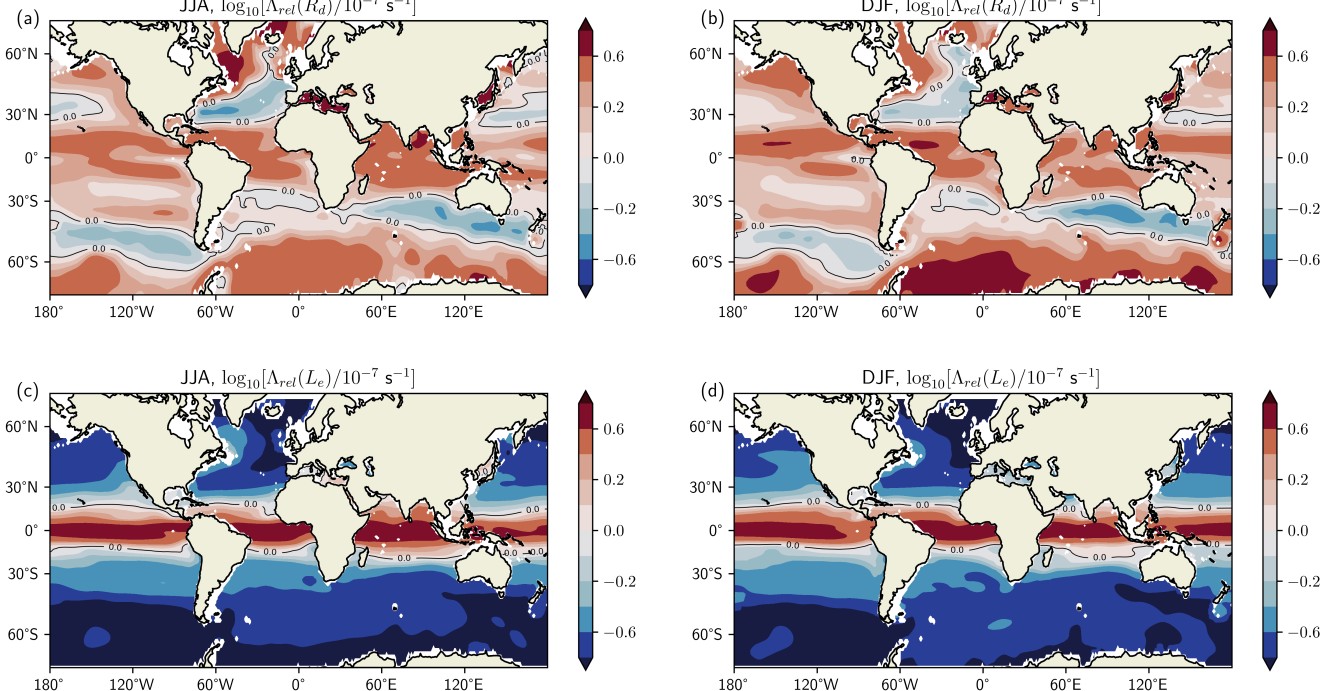

**Figure 8.** A global dissipation rate for relative wind stress damping, $\Lambda_{rel}$ for: left) JJA; and right) DJF. In a,b) $\Lambda_{rel}$ is a function of $R_d$, and c,d) $\Lambda_{rel}$ is a function of $L_e$. The dissipation rate is normalised by a constant dissipation rate, $10^{-7}$ s$^{-1}$ used in Mak et al. (2018). It is then shown on a $\log_{10}$ plot. The colorbar has uneven intervals, with smaller steps around zero to highlight when both dissipation rates are equivalent, or $\Lambda_{rel}$ is marginally less than or greater than $10^{-7}$ s$^{-1}$. The thick contour line represents the point where $\Lambda_{rel} = 10^{-7}$ s$^{-1}$.

compared with those in the Southern Ocean. In the equatorial and tropic regions, $\log_{10}(\Lambda_{rel}/10^{-7}$ s$^{-1})$ is mostly positive with

contributions from wind speed and reduced gravity. Across the seasons the spatial pattern in $\log_{10}(\Lambda_{rel}/10^{-7}$ s$^{-1})$ is similar, with only minor differences arising from changes in $\boldsymbol{u}_a$, $\mu$, $g'$, and $R_d$.

Figure 8c,d shows the dissipation rate that depends on the e-folding length scale, $L_e$. We see that $\log_{10}(\Lambda_{rel}/10^{-7}$ s$^{-1})$ is largely negative, except over the equatorial band, where $L_e$ is smaller than $R_d$ here, increasing the dissipation rate. Throughout the Southern Ocean and western boundaries, we find that $\Lambda_{rel}$ is around a tenth to a quarter the size of $10^{-7}$ s$^{-1}$. We also see

that the patterns are similar to those seen in $\Lambda_{rel}(R_d)$ (Fig. 8a,b), since the spatial pattern of the chosen eddy length scale ($R_d$ or $L_e$) does not vary, as $L_e$ depends on $R_d$.

Contrasting the two choices of eddy length scale is summarised using a density plot of $(\Lambda_{rel}/10^{-7}$ s$^{-1})\hat{lon}$ in Fig. 9. Here, we have weighted $\Lambda_{rel}/10^{-7}$ s$^{-1}$ with a normalised longitude ($\hat{lon}$), where the largest weight is at the lowest latitude. Overall, the distribution of the dissipation rate is consistent with the spatial plots seen in Fig. 8. The density of dissipation rates

depending on $L_e$ are skewed to the left and exhibits a narrow range centred around 0.2. The density of the dissipation rate



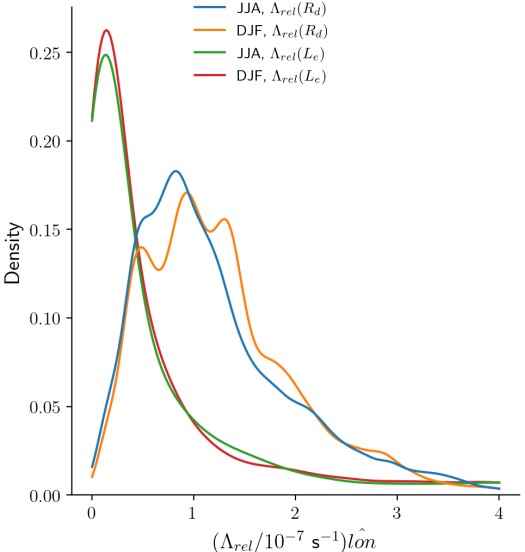

**Figure 9.** Kernel density estimation of $(\Lambda_{rel}/10^{-7}\ \mathrm{s}^{-1})\hat{lon}$, where $\hat{lon}$ is a normalised longitude. The four lines represent the dissipation rate $\Lambda_{rel}$ depending on the chosen eddy length scale and season (DJF or JJA).

depending on $R_d$ is shifted to the right and displays a wider range of values centred close to 1. What Fig. 9 shows is that the dissipation rate due to relative wind stress may lie somewhere between $2 \times 10^{-8}\ \mathrm{s}^{-1}$ and $4 \times 10^{-7}\ \mathrm{s}^{-1}$.

## 6 Summary and discussion

In this work we have presented a constrained eddy energy dissipation rate for a well-known and important mesoscale dissipation

pathway, relative wind stress. Deriving this dissipation rate draws on our fundamental understanding of relative wind stress damping, vertical eddy structure, and eddy energy. The intention with this dissipation rate is for it to fit into an existing eddy energy budget-based eddy parameterisation (e.g. GEOMETRIC), and offer improvements to the relatively unconstrained and spatially homogenous dissipation rate currently employed.

Before the proposition of a dissipation rate, an approximate expression for relative wind stress damping, termed $P_{rel}$, was

found in Section 2.1. Several assumptions were made to help achieve this expression found in Eq. (11): mesoscale eddies are, on average, Gaussian in shape over the global ocean (Chelton et al., 2011); and the wind field is constant in strength and direction (Duhaut and Straub, 2006). Thereafter, $P_{rel}$ is used to predict the decay of baroclinic eddy energy in a analytical two-layer model, which is described in Section 2.2. The analytical model is chosen to represent a mesoscale eddy with a first baroclinic mode structure, consistent with the first baroclinic mode containing a high portion of eddy energy. Then, comparing

the evolution of eddy energy in the analytical model with a general circulation model shows that the expression for relative wind stress damping can approximate the decay of eddy energy well in each eddy type for around 150 days (Fig. 3). However,





it is important to highlight that these results are dependent on our choice of model parameters. For example, modifications to eddy parameters could alter total eddy energy, relative wind stress damping, and instability timescales. Nevertheless, we would still expect damping by winds to be the same across each model due to the matching of eddy amplitude.

The key component of this work lies in the proposed dissipation rate for eddy energy due to relative wind stress, outlined in Section 5. The dissipation rate $\Lambda_{rel}$ culminates from the theory given in Section 2 and the verification of $P_{rel}$ through the use of a general circulation model in Section 4. Deriving the dissipation rate $\Lambda_{rel}$ in Eq. (28) is based on a simple two-layer analytical model that exhibits a first baroclinic mode structure. This model is chosen because the eddy sea surface height reflects the movement of the first baroclinic mode, and can as such represent a large portion of eddy energy (Chelton et al.,

1998). An analytical expression for total eddy energy $E$ is then calculated from the two-layer theory. From this, we are able to construct an eddy energy dissipation rate due to relative wind stress, $\Lambda_{rel} = P_{rel}/E$. This dissipation rate is assumed to depend on available potential energy in the thermocline, and not kinetic energy. So whilst relative wind stress damps the surface geostrophic motion, the greater dynamical impact is for relative wind stress to relax the eddy thermocline displacement, and damp potential energy.

A global map of the dissipation rate is presented in Sect. 5 along with the terms that contribute to it. The eddy length scale is considered to be either the first baroclinic Rossby radius of deformation ($R_d$), acquired by solving a typical eigenvalue problem, or an e-folding length scale ($L_e$), computed using data from Chelton et al. (2011). The two eddy length scales help to form a a range of values that the $\Lambda_{rel}$ could take. The dissipation rate $\Lambda_{rel}$ is shown in Fig. 8 normalised by a constant dissipation rate $10^{-7}\,\mathrm{s}^{-1}$ on a $\log_{10}$ plot. For $R_d$, we find that $\Lambda_{rel}$ is greater than $10^{-7}\,\mathrm{s}^{-1}$ across much of the ocean, with

hotspots throughout the Southern Ocean, tropics, and equatorial regions. In the western boundary currents, $\Lambda_{rel}$ is closer to $10^{-7}\,\mathrm{s}^{-1}$. For $L_e$, $\Lambda_{rel}$ is less than $10^{-7}\,\mathrm{s}^{-1}$ over most of the ocean except the equatorial region. However, $\Lambda_{rel}$ still makes up to a quarter of $10^{-7}\,\mathrm{s}^{-1}$ in regions like the Southern Ocean and western boundaries. Enhanced eddy energy dissipation in the Southern Ocean could impact heat and mass transport (Meijers et al., 2007; Stewart and Thompson, 2015), the exchange of heat and carbon at the air-sea interface (Villas Bôas et al., 2015; Pezzi et al., 2021), and Antarctic ice cover (Munday et al., 2021).

The values in the western boundary currents are much less pronounced than in the Southern Ocean, hinting at the regional variation in eddy energy dissipation (Mak et al., 2022b). Seasonal variations are present in the dissipation rate, particularly in eddy rich regions, and are consistent with changes in wind speed. High frequency wind events can also take place (Zhai et al., 2012), which may significantly modulate eddy energy dissipation in some regions.

The dissipation rate is based on a linearised model which, by design, neglects many phenomena that take place in the

ocean, such as instabilities and wave dynamics. In the time evolution of total eddy energy (Fig. 3), the predicted and MITgcm results were shown to diverge around day 150 in each eddy type. Total eddy energy in MITgcm was found to undergo an exponential like decay for around 20 days, which corresponded with a change in eddy shape (Fig. 4). A foundation of the linearised prediction method assumes that the baroclinic eddy remains circular, however, this is clearly not the case. The MITgcm eddy begins as a coherent structure, and then transitions into two smaller eddies. The splitting of a baroclinic eddy is

due to baroclinic instability and leads to the formation of two barotropic eddies via barotropization (Ikeda, 1981; Dewar et al., 1999). This suggests that our predictive method could benefit from including an additional model that accounts for a smooth

transition to the two smaller barotropic eddies. Indeed, the timescale for this transition could depend on a baroclinic mode timescale, and might even depend on eddy polarity. Whether accounting for this process in this prediction method is important for long climate timescales is something that could be investigated in a future body of work.

This study presents a constrained eddy energy dissipation rate due to relative wind stress damping. Although relative wind stress is not the only mechanism associated with eddy energy dissipation, its focus in this study is grounded in the effects it has on ocean dynamics and ocean processes (Seo et al., 2016; Wu et al., 2017; Renault et al., 2019). What is more, having a simple analytical expression for this dissipation rate, which can then utilise ocean datasets is a further advantage to this work. Being able to then illustrate the global variability in the eddy energy dissipation rate due to relative wind stress enables the discussion

of possible implications this could have on wider climate processes. Areas of immediate future work should look to determine a reasonable approximation for eddy length scale and examine the impacts of this dissipation rate in a global ocean model. Furthermore, we hope the work here could provide the basis for similar studies looking to constrain an eddy energy dissipation rate, improving the energetics and flow in global ocean models.

*Code and data availability.* NCEP-NCAR Reanalysis 1 wind speed data provided by the NOAA PSL (NOAA PSL, https://psl.noaa.gov/data/
gridded/data.ncep.reanalysis.html, last access: 10th November 2022). Temperature and salinity data is from World Ocean Atlas provided by NOAA NCEI (NOAA NCEI, https://www.ncei.noaa.gov/access/world-ocean-atlas-2018/, last access: 10th November 2022). The remaining data, including the computed dissipation rates, and code to reproduce the results in this work can be found at doi.org/10.5281/zenodo.8017212.

*Author contributions.* All authors contributed to the conception and design of this work. TW worked on the analytical derivations and its numerical solution, idealised model design, formal analysis, figure production, and writing - original and review. XZ provided supervision
of the work, administered the project, assisted in solving the eigenvalue problem, and writing - review and editing. DM provided supervision of the work, assisted in the analytical work and MITgcm setup, and writing - review and editing. MJ provided supervision of the work, and writing - review and editing.

*Competing interests.* The authors declare that they have no conflict of interest.

*Acknowledgements.* This work was supported by the Natural Environment Research Council through the EnvEast Doctoral Training Part-
nership (Grant NE/L002582/1). TW thanks XZ, DM and MJ for their guidance and mentorship throughout this work. The authors would also like to acknowledge the computing resources provided by the University of East Anglia.



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
