# Peer review of "Constraining an Eddy Energy Dissipation Rate due to Relative Wind Stress for use in Energy Budget-Based Eddy Parameterisations"

_EGUsphere, 2023_

## Author Comment (AC1)

**Reviewer 1 response:**

The authors would like to thank the reviewer, Dr. Julian Mak, for their constructive and creative feedback on this manuscript. Below, we address each point in order.

Technical comments:

- There is the paper of Rai et al. (2021) [Rai et al., Sci. Adv. 2021; 7 : eabf4920] that looks into eddy killing, providing an estimate of energy loss from the ocean arising from relative wind stress. The paper is close enough in spirit that appropriate citations should be made, probably in multiple places throughout the text, and in particular highlighting how the present methodology is different. Would be good to see a comparison on how the present results are similar/different to the results there.

  (There is also a Rai et al (submitted) but that's not published yet, and may be not as relevant as the 2021 paper.)

  > *Response:*
  > Thank you for pointing us in the direction of this study. To start, the intention of our study is to produce a simple dissipation rate and give a flavour of what it could look like. Rai et al. make some important conclusions regarding the length scale of eddy killing and its seasonality i.e. peaking in winter. They base this on the seasonality of EKE. In our work, the eddy dissipation parameterisation is based on the dissipation of APE, since this is typically the largest reservoir of eddy energy.
  >
  > We have cited Rai et al. (2021) in multiple places throughout where appropriate.

- A two-layer shallow water model is employed as a way to have a representation of the first baroclinic mode, however the choice of "mode" (or the basis) is the "standard" choice assuming flat bottom essentially. That choice of basis is not unique and increasingly there are studies using surface modes (e.g. LaCasce 2017; Groeskamp et al. 2020; Ni et al., accepted at GRL or 2023?) I am not suggesting you redo your analysis using surface modes, but given there is a choice of basis, minimally it would be good to speculate how the results are going to be the same and/or different (although of course if you redo the analysis, instead of speculating you could then quantify the similarity and differences, and strengthening the conclusions).

  > *Response:*
  > Thank you for bringing this point to our attention. We feel that the flat bottomed baroclinic mode suits our experiment well, particularly with regards to our two-layer baroclinic model and MITgcm setup.
  >
  > We include some sentences on surface modes on lines 148-149 and 483 with relevant

> citations.

- Regarding Eq. (19), I'll be honest and didn't think about this as much as I probably should have. Two points here:

  1) The notation is a little un-rigourous, because it's not clear which quantities are scalars, vectors and/or tensors, missing some contraction operators and the like. For example, the final term on the right hand is written $(\nabla^2 u)^2$, but you really mean $\nabla^2 u \cdot \nabla^2 u = |\nabla^2 u|^2$. Is the second term a tensor ($|\nabla u|^2$) the hit by the Laplacian operator? This is mostly notation but probably could be cleaned up a bit.

  2) Normally in shallow water it is known that simple choices of the Laplacian as a diffusion leads to sign indefinite energy dissipation (e.g. Peter Gent in 1993, "The Energetically Consistent Shallow-Water Equations"; Gilbert et al., 2014, "On the form of the viscous term for two dimensional Navier–Stokes flows"; in my PhD thesis). One offending reason is that the primitive variables in shallow water are (h, u, v), but the conservative variables are (h, U = hu, V = hv). If for example in the prognostic equation we have $-\nabla^2 u$, then multiplying by *hu* then integrating by parts gives (abusing notation a bit)

     $\int hu \cdot \nabla^2 u = boundary\,terms - \int \nabla(hu) \cdot \nabla u,$

     which gives the expected sign-definite term $-h|\nabla u|^2$, but there is a cross term involving $\nabla h$ floating around. The problem however doesn't exist in the conservative variables. It seems somehow you don't have this issue here? I am not seeing how your terms could be written as a flux given there is that annoying $h$ term floating around, so a clarification would be useful.

     Ultimately I assume it's quantitatively not going to be important, because your "hyperdiffusion" $\nabla^4 u$ (I put quotation marks because I am arguing $u$ is not the variable you should be hitting $\nabla^4$ with) is presumably going to be a small effect. Clarification and maybe appropriate references here would be helpful (e.g. Gent 1993; note also he makes the point about energetic consistency, which I am not convinced you have here, but it's probably not too important).

> *Response:*
> Thanks for pointing this fact out. On reflection, the viscous term is not overly important and so we will leave it out of the paper entirely. The text in sections 2 and 4 have been modified accordingly and Fig. 3 has been changed too to account for zero damping in the absolute wind stress cases.

- So intuitively the action of the wind is at the surface, but that is being argued to be felt over the structure of the baroclinic mode, so there is implicitly some assumption about the time-scale of communication. Do we actually know what that is, and is that small enough for the conclusions here to internally consistent? I think I wouldn't raise this question so much if surface modes were used for example. Clarifications on this point would be welcome, maybe commenting on the evolution of the MITgcm model results.

> *Response:*
> Thank you for this point. We felt adding a comment directly in the results of the MITgcm results (Section 4) didn't feel right. So, we have added a few sentences at the beginning of section 5, line 335-338. We essentially add a further justification (relative wind stress timescale of communication) for using potential energy, rather than just a scaling and that potential energy is largest.

- The toy model uses an eddy as a noun (quasi-circular object), but results are then applied generically to a velocity with no specific mention to the eddy, so I assume there is no eddy detection that is being done here, and the application is then considering eddies as the verb (the fluctuation, a special case being the quasi-circular object). Could the authors comment on this distinction, and which one would be the more appropriate thing to do? One could for example argue that we might want to identify the coherent eddies (so the noun, via Eulerian or Lagrangian approaches), then consider the impact of relative wind-stress on those identified eddies for consistency of the theory and application.

  (I don't personally believe you should do what I just suggested. I am just raising the point that the word "eddy" is sometimes used to mean different things by different people, and sometimes the intention is not as precise as it should be.)

> *Response:*
> You are right. We initially consider the eddy to be a noun in the toy model, and this helps us derive the parameterisation. In our application of the parameterisation, we are considering eddies to be deviations from the mean, of which coherent eddies make up a small part of this. We have added the following two sentences on line 347-348 "In addition, we now consider *eddies* to be deviations from the time-mean, rather than just being a singular coherent eddy. The interpretation of the dissipation rate can also be thought of as one for these eddy time-mean deviations."
>
> We have also added a sentence on line 469 that suggests the use of a eddy detection method in future work.

- Maybe a hypocrite for asking this (because I didn't do it either in the 2022a paper), but a dissipation rate is estimated here but no estimation of a power? Could you estimate a

power, and how might that compare with the results of Rai et al (2021) say? Or if you refrain from doing so, give a reason on why you don't?

> *Response:*
> Computing an estimation for power is not something we considered doing. We could try and attempt an estimation, but this would require eddy detection e.g. eddy amplitude $A$ and eddy radius $R$, similar to Chelton et al. (2011). We would put the detected values of $A$ and $R$ into our equation for $P_{rel}$ and compute globally. This is something we could do, but is beyond the scope of the papers goal. Again, perhaps an idea for a future piece of work.

Presentation comments

- (line 6): unless you can definitively say and show the "nonlinear baroclinic processes" causality, I would lessen the strength of the wording and say "there is divergence from the analytical model at around day 150, likely due to the presence of nonlinear baroclinic processes" (because it's really more an observation at the moment).

> *Response:*
> Agree. Changed.

- (line 12 and later): the $10^{-7}$ is mentioned but its significance (or lack of) is never given explicitly. Easiest to say up front why that that reference value is chosen.

> *Response:*
> Removed $10^{-7}$ for now in abstract.

- (two sentences spanning line 10-12): reads a bit clunky, could do with a re-write.

> *Response:*
> Sentence changed/combined.

- (line 34-35): I would argue that's not a good comparison, because the low explicit eddy energy is to do with the coarse resolution model and much less on the GM parameterisation itself. I would personally just remove that sentence.

> *Response:*
> Removed.

- (line 45-47): Jumpy sentence, consider rewrite (eddy saturation <-> GEOMETRIC while "other" <-> turbulent energy cascade, and as written the is ambiguity in how the sub-clauses are related to each other)..

> *Response:*
> Removed the Jansen citation and focused on GEOMETRIC.

- (line 59): formatting of reference, brackets.

> *Response:*
> Fixed.

- (paragraph of line 60): Rai et al. (2021) should be cited and results compared accordingly in this paragraph.

> *Response:*
> Included on line 69.

- (line 95): $g$ should be the gravitational acceleration constant

> *Response:*
> Done.

- (line 95): "$\nabla_h$ IS THE horizontal gradient operator" or simila

> *Response:*
> Done.

- (sentence of line 114-115): Could be read as there is current feedback onto the wind profile, which I assume is not what was intended.

> *Response:*
> Not what I mean. Have removed that last sentence and included some description of what the figure shows in the text.

- (equation 9): consider using \begin{align} \end{align} with some & according to break the lines, probably

```
\begin{align}

    W_{rel} &= \tau_rel \cdot u_g \\\\

            &= etc. \\\\

            &=

\end{align}
```

> *Response:*
> Done.

- (line 143-145): citation to Rai et al. (2021) here also probably.

> *Response:*
> Done.

- (section 2.2 opening paragraph): probably comment about surface modes here, or say the discussion will be given in the conclusion section

> *Response:*
> Included LaCasce citation but discuss in conclusion.

- (line 163): "...where $\cdot_{1,2}$ denotes the upper and lower layer variables,..."

> *Response:*
> Done.

- (line 202, 203): might consider swapping $u$ and $\eta$ ordering so you can have equation reference ordering as (5) and (6)

> *Response:*
> Done.

- (line 223): not sure why you wouldn't just RK4 the whole thing, comment to clarify? (Because piggy-backing on the MITgcm AB3 time-stepper?)

> *Response:*
> Clarified.

- equation (23): \left( and \right) for brackets

> *Response:*
> Done.

- (liner 262): so you have sponge layers or diffusion to soak it up? clarification would be useful (if discussed in previous work, citation here would also be appropriate).

> *Response:*
> This information is provided in Wilder et al. (2022) and first cited in the first paragraph of section 3.1.

- (line 273): I might have opted to define $n_0(z)$ as the NEGATIVE vertical gradient to soak up that negative sign in *PE* partly because it is never used again anyway (I'm not remotely attached to this suggestion).

*Response:*
Thanks. Will leave as is though.

- (line 318): remove comma maybe, "...but with vertical diffusion did result..."

*Response:*
Done.

- (line 343): "...zonal wind velocity, because wind patterns..."

*Response:*
Done.

- (line 345): "...is on a 2 degree horizontal grid."

*Response:*
Done.

- (line 355): comment on differences / similarities with Rai et al. (2021)

*Response:*
Added a sentence on line 392-393 when describing figure 7.

- (text below equation 31): comment on how surface modes might change results here, or say this will be talked about in conclusions section (I would minimally speculate what is expected to change with surface modes here though).

*Response:*
Comments on surface modes will go in the discussion. We feel that they will flow better in an 'alternative approach' paragraph at the end. See line 483.

- (line 370-372): "...while there is a slow down...between seasons, with the largest absolute values..."

*Response:*
Done.

- (line 388): against, may want to be explicit about significance of 10^-7

> *Response:*
> Done.

- (line 390): extra space after $0°$

> *Response:*
> Done. Added a dash after.

- (line 395): for completeness it is also in Mak et al (2022a), Fig 7 I suppose.

> *Response:*
> Added other citation here.

- (line 405): is negative dissipation a problem?

> *Response:*
> There is no negative dissipation? We take the log of $\Lambda_{rel}/10^{-7}$.

- (line 407): would recommend $\widehat{lon}$, or even better $\widehat{\text{lon}}$

> *Response:*
> Done.

- (line 428): how significant are the uncertainties? a quantification would be useful

> *Response:*
> It's difficult to quantify really without diving into more experiments. Have modified the sentence on line 445.

- (line 438-439): is the issue of "time-scale of response" an issue?

> *Response:*
> It shouldn't be. See our earlier point in technical comments.

- (line 451): Mak et al (2022a) the one actually intended? (2022b is the prognostic calculation, while the 2022a is the inverse calculation). How are the results here compared to that say?

> *Response:*
> Removed sentence. Didn't really follow from the previous sentences.

- (line 467-468): Rewrite? I assume you want "A further advantage of this work is having a simple analytical expression for this dissipation rate that can be applied to ocean datasets" or something similar. It doesn't flow very well at the moment.

*Response:*
Thanks.

---

## Author Comment (AC2)

**Reviewer 2 response:**

The authors would like to thank the reviewer for their insightful and constructive feedback on this manuscript. Below, we address each point in order.

Section 5:

1. Line 367: There is a formula given here for the reduced gravity, g'. The first problem I have with this formula is the term involving the eigenmode, phi_1. The eigenmodes can always be scaled by a constant multiplying factor and since we are not told how the eigenmodes are normalized, the formula for g', as written, is not sensible. It follows that we need to be told how the eigenmodes are normalized. Nevertheless, I cannot make sense of where this expression for g' comes from. The fact the value of the eigenmode at z=0 appears is mysterious; the fact it appears squared even more so. Some explanation is required.

> *Response:*
> The equation for $g'$ is correct, although we accept the description could be made clearer. In Xu et al. (2011), to get Eq (37) on page 539, they rearranged (from table 2 page 352 in Flierl (1978)) $F_1(0)$ for $H_1$, which is where the square of the eigenmode and its definition at $z = 0$ comes from. From the same table in Flierl (1978), the eigenvalue $\lambda_1$ is rearranged for reduced gravity $g'$, where $g' = \varepsilon g$ (see Flierl, page 347). Then substituting (37) into (38) from Xu et al. (2011), the equation for $g'$ follows. In the updated manuscript we keep $g'$ in terms of $\lambda_1$, and not $c$. So, $g' = \frac{f^2(1+\phi_1^2(0))}{\lambda_1 H_2}$, where $\phi_1$ is normalized using Eq (36) in Xu et al. (2011).

2. In the 2-layer model, the wave speed, c, for the first baroclinic mode is related to the reduced gravity by c^2 = g' H_1 H_2/H where H = H_1 + H_2 and H_1, H_2 are the undistiurbed depths of the upper and lower layers respectively (see Gill(1982), equation (6.3.7)). Presumably the term involving the eigenmode in the expression for g' above corresponds to the factor H/H_1?

> *Response:*
> Yes, it does. In the previous response, when rearranging to find $g'$, we actually have the equation $g' = \frac{c^2 H}{H_1 H_2}$. But in our equation, we replace $H$ with $H_1(1 + \phi_1^2(0))$, and do not use $c$.

3. Line 366: Here, it is written that H_1 is taken to be the depth of the zero crossing of the eigenmode. Why write this in terms of the minimum of the absolute value of the eigenmode when this is obviously zero at the zero crossing? I also wonder if this is the appropriate choice for H_1? It seems reasonable but might also turn out to be on the deep side? On the other hand, it is clear that the choice of H_1 is of great importance

since once H_1 is known, so is H_2 and hence g', given that c and H are known (see above).

> *Response:*
> This is a mistake. We will write "We take the depth of zero crossing of the first baroclinic mode to be $H_1$" on line 375-376. We did also attempt to find $H_1$ by computing the depth of the main thermocline, but discovered this to be a non-trivial task. We therefore chose the depth of zero crossing of the first baroclinic mode as $H_1$.

4. Line 369: How is mu computed? We are not told anything about this. On line 186, mu is given by mu = -g'H_2/gH. How does this connect to the mu being used here?

> *Response:*
> $\mu$ is computed using the expression on line 185. We have reiterated this fact in the text on line 378.

5. I found myself wondering if a simpler model for an eddy than the 2 layer model used here (i.e. allowing the lower layer to be in motion and selecting only the baroclinic model) might be to assume that the lower layer is at rest, corresponding to a projection onto both the baroclinic and barotropic modes. Such a set-up would be equivalent to the 1 ½ layer model. The 1 ½ layer model is recovered from the 2 layer model in the limit that the depth of the lower layer, H_2, goes to infinity. The mathematics for the 1 ½ layer model is simpler than for the 2-layer model and I feel sure the expression in equation (29) would be the same as given in the manuscript with mu = g'/g (since H_2/H tends to one as H_2 tends to infinity). This shows very clearly the importance g' for determining the dissipation rate. The authors might want to consider briefly discussing the 1 ½ layer model as an alternative to their single baroclinic mode setup.

> *Response:*
> The 1.5-layer model is certainly interesting and offers an alternative approach to the 2-layer model. We have briefly mentioned how the 1.5-layer model alters the dissipation rate in Eq. (28) (old Eq. (29)), showing the importance of the reduced gravity $g'$. The choice of the 2-layer model in this work essentially follows on from some earlier work we carried out on the 1-layer model (see this thesis https://ueaeprints.uea.ac.uk/id/eprint/92511/).

6. An obvious question that is not addressed is what the authors do about the fact the ocean actually has variable bottom topography. I assume that the eigenvalue problem stated at the beginning of Section 5.1.1 uses H = 4000m? This should be made clear. How do the authors deal with the temperature and salinity data if the local depth is less than 4000m? Again, this needs to be explained.

> *Response:*
> Thanks for pointing this out. The eigenvalue problem uses the depth from the WOA dataset, so we have made this clear on line 371. We also neglect data shallower than 300 m, so we write this on line 373. Reviewer 1 (Julian Mak) suggested the use of surface modes which account for rough bathymetry. This may be something that we should look at in future work.

7. Another issue throughout section 5 and in the figures is how the authors deal with the Rossby radius of deformation near the equator where the Coriolis parameter goes to zero? I assume a band around the equator must be left out. But we do not see this in Figure 7 and neither is anything ever said about it in the text.

> *Response:*
> Thanks for bringing this up. We did not originally deal with the Rossby radius of deformation at the equator. Considering this fact, we therefore include masks over the equator regions in the Rossby radius (Fig. 7), and the dissipation rate (Fig. 8). The density plot (Fig. 9) has also been recomputed by neglecting the equator band, and the figure is not too different to the previous one. The mask is pointed out in the figure captions, and in text on lines 393-394 and 414-415.

8. Line 334: We need to be told what expression is being used for the energy in (29). In particular, we need to be told which term in (17) comes from the thermocline displacement? I also feel confident that it is easy to show that this term dominates the expression in (17) by substituting appropriate values for the parameters. Showing this would justify the use of the "key assumption" made here and make it clear it is not an ad hoc assumption.

> *Response:*
> This has been made much more clear in the text. We have included a scaling and have cited Gill et al. (1974) for the dominance of potential energy. A simplified eddy energy $E$ has now been shown [see new Eq. (27)]. We have removed the phrase "key assumption" because it is rather a fact.

9. It seems that the wind speed used in (29) is the seasonal mean wind speed. This also needs to be stated clearly. What about the fact that the winds are not steady? How could this affect the result? At least in (29), wind speed appears linearly (although the speed itself is not linear). In reality, however, the drag coefficient also depends on the wind speed. Indeed, I am reminded of the paper by Thompson, Marsden and Wright (1983, JPO) from 40 years ago...

> *Response:*
> In Section 5.1, we state that the data is averaged into seasons (line 356). On line 468-469 we briefly mention the work by Zhai et al. (2012) on high-frequency wind power input, where high-frequency wind may also impact the dissipation rate.

10. Line 385: Is L_e being used or what is written on line 355?

> *Response:*
> Thanks. Will make clear that we are using either $R_d$ or an eddy length scale $L_e$ scaled by $R_d/\hat{R}_d$.

11. Lines 389-390: To talk about the influence of bottom topography is a bit glib since what is plotted comes from equation (29) and hence must arise from the terms in (29). Maybe R, maybe the wind speed? Of course, it is true that this is where the Antarctic Circumpolar Current takes a turn to the north, consistent with the form drag effect across the Drake Passage sill, so bottom topography may play an indirect role.

> *Response:*
> In this text we are trying to connect our results with something tangible that readers can understand more easily. Indeed, the results are a result of $R$ or wind speed, but $R$ is also directly a result of temperature and salinity, which must be steered to some extent by features in the Southern Ocean.

12. Line 394: Likewise, here, the authors could (should) be able to say which terms in (29) are making the important contribution.

> *Response:*
> Yes. Have added more detail to line 407-409.

13. Line 409: The dissipation rates plotted in Figure 9 have to be consistent with Figure 8 by construction!

> *Response:*
> Removed the line.

Other comments

1. Lines 20-21: There are lots of examples of how eddies modulate volume transport, going back to the early quasi-geostrophic models of Holland et al., e.g. Holland, Rhines and Keffer (1984), or even Holland (1978), to more recent papers such as Wang et al. (2017, GRL.

> *Response:*
> Thanks. We have included Holland (1978) and Wang et al (2017) on line 21.

2. Lines 33-35: Tandon and Garrett (1996, JPO) were perhaps the first authors to ask what happens to the energy released by the GM scheme. The discussion here reminds me of the backscatter scheme...

> *Response:*
> Thanks. We have included this citation on line 33

3. Line 37: Although Jansen et al. gets referenced, there is no explicit mention of the backscatter scheme in the text?

> *Response:*
> That's right. The point we want to make with that reference is that they use an energy equation to inform a GM transfer coefficient, and thus constraining the dissipation of energy is important for these types of parameterisations.

4. Line 63: The drag coefficient is known to be a function of wind speed – there is no "could be" about it! Think of Large and Pond (1981, JPO) and all the subsequent updates.

> *Response:*
> Corrected.

5. Equation (6) should not include "A" explicitly since the amplitude is already contained in eta as given by (5).

> *Response:*
> Corrected.

6. Equation (8): It should be noted that this approximation assumes that that the absolute value of u_a is much bigger than the absolute value of u_g.

> *Response:*
> Have added a sentence to line 109

7. Equation (10): I assume that the drag coefficient is assumed to be a uniform constant here, independent of u_a?

> *Response:*
> Yes. Have added a sentence in section 2.1.2 after Eq. (7) on line 105 saying $C_d$ is kept

> constant.

8. Equation (10): Furthermore, if I understand correctly, it is not W_rel that is being integrated but only the last 4 terms in the expression for W_rel in (9c)?

> *Response:*
> Yes, but for consistency sake in text we write the integral of $W_{rel}$.

9. Line 142: It is already clear from the last four terms in (9c) that P_rel is negative. This is because the only term that is not negative-definite is the (u_g)^3 term and this integrates out for a circular eddy. I am also noting here that the absolute value of u_a, plus u_a itself, is always positive or zero.

> *Response:*
> Yes you are right. It's interesting to look at the terms individually, but we will leave the discussion as is.

10. Lines 151-152: Better to write "zero net vertically-integrated flow".

> *Response:*
> Done.

11. Line 163: Should say that eta_2 is measured positive upward. This is because interface displacement in the 2 layer model is often measured positive downward.

> *Response:*
> Thanks. Added.

12. As I mentioned earlier, the authors could consider also discussing the simpler 1 ½ layer model in addition to their 2 layer model.

> *Response:*
> Thanks. We have included a brief mention of the 1.5-layer model on line 343-346.

13. Line 186: It should be noted that these solutions for lambda and mu are only valid in the limit g'/g goes to zero – see Gill (1982), Section 6.2. From (13), for the barotropic mode this is obvious because lamda = 1 implies g' = 0. But it is also true for the baroclinic mode.

> *Response:*
> Done.

14. Lines 198-199: It is not correct to say that the terms in the middle represent the redistribution of energy by nonlinear advection. The pressure work term is also contained in these terms! Think of the energy equation for the equations linearized about a state of rest, e.g. Gill (1982), Section 5.7.

> *Response:*
> Added a line on 196 to add 'pressure work terms'.

15. Line 215: Surely the higher order boundary conditions on biharmonic viscosity must be used for the first two terms on the right hand side of (19) to drop out on integration? The no normal flow condition is not enough on its own.

> *Response:*
> Have removed the viscous term. See comments/response by/to reviewer 1.

16. Section 3.1: I am assuming that the imposed atmospheric wind is uniform, that the drag coefficient is independent of wind speed and that the model domain has closed boundaries? Please be clear about these things! Also, what depth is used for the model ocean, 4000m?

> *Response:*
> As we mentioned at the beginning of the section, more details are given in Wilder et al. (2022). Have included ocean depth in the text. The default drag coefficient in MITgcm depends on wind speed and this is what we use. At moderate wind speeds we don't expect there to be too many differences in wind power input (see this thesis on page 55 https://ueaeprints.uea.ac.uk/id/eprint/92511/). Have added a small paragraph from line 252 detailing the wind field.

17. Line 260 and thereabouts: It might be helpful to show a plot of the vertical profile of the eddy used for initialization?

> *Response:*
> We don't show that here since we showed that in Wilder et al. (2022).

18. Section 4: I do not like the title. How about "Verifying the analytic model"?

> *Response:*
> That's a better title, thanks.

19. Section 4, first paragraph: It should be stated in the text that this paragraph refers only to the first 150 days of integration, otherwise the impression one gets from Figure 3 is quite different.

*Response:*
Have added 'Now, focusing on the first 150 days...' on line 291.

20. Lines 318-319: What is the vertical mixing scheme used in the model? Vertical mixing by itself will always increase potential energy – vertical mixing does work against gravity by mixing dense water upwards. Perhaps the greater increase in potential energy due to vertical mixing in the cyclonic case is because the stratification is weaker than in that case?

*Response:*
The model uses only an explicit vertical diffusion and viscosity (see Wilder et al. (2022)). Examining the impact of vertical mixing on eddy energy is something that we could look to examine further at a later date.

21. Figure 4: I assume what is plotted is geostrophic relative vorticity, not total relative vorticity? Please make clear.

*Response:*
Added to caption.

22. Line 454: The 2 layer model used by the authors is not "linearized", as written in the text, although the nonlinear terms do not play a role in the analysis.

*Response:*
Thanks. Have changed the start of this paragraph slightly.

---

## Author Response (AR2)

Many thanks for these additional comments, we appreciate the time and effort you have put in to improve this manuscript.

**Reviewer 1**

Minor comments:

- line 10: "found to vary from 0.25 to 4 times"?

> *Response:*
> Done.

- line 23: being picky here, there is no eddy dynamics as such in a non-eddy resolving model, and "eddy feedback" onto the mean state or similar is really what is being meant

> *Response:*
> Modified.

- line 25: capital W in "McWilliams". In bibtex, can do this by author = {P. R. Gent and J. C. {McWilliams}}

> *Response:*
> Fixed.

- line 30, 31, eq(1): inconsistency with kappa or kappa_gm, choose one (probably latter)

> *Response:*
> Sorted.

- line 50 (and line 72-73): either acronym used before being defined (ACC + AMOC), or be consistent and use AMOC in lin 72-73 (probably the former)

> *Response:*
> Sorted.

- line 50: could have a reference to Marshall et al (2017) for the mechanism that increasing eddy dissipation increases transport

> *Response:*
> Added prior to Mak et al. (2022b).

- line 55: "numerical optimisation" could mean optimising a numerical model, which is definitely not what was done in that paper. "Inverse method" or "PDE-constrained optimisation" would be more unambiguous

> *Response:*
> Added.

- eq 3: no comma

> *Response:*
> Done.

- line 153: Wunsch (1997) to be more specific and consistent with prose style.

> *Response:*
> Done.

- line 250: would think that "close to zero" is sufficient. I guess "close to, but not, zero" would be ok, but that seems unnecessarily jumpy

> *Response:*
> Done.

- line 252: consider "background wind; see Wilder et al (2022)" since the two clauses are closely related

> *Response:*
> Done.

- line 291: "We first focus on the first 150 days"

> *Response:*
> Done.

- line 394: full stop at end of sentence

> *Response:*
> Thanks.

- line 404: unnecessary dash after 0 degrees

> *Response:*
> Fixed.

- bib: the Gent and McWilliams reference typesetting as above

*Response:*
Fixed.

Many thanks for these additional comments, we appreciate the time and effort you have put in to improve this manuscript.

**Reviewer 2**

Minor comments

1. Line 25: it is not correct to say that the GM scheme "advects tracers downgradient". In fact, for most tracers there is no connection between the GM advective velocity and the tracer concentration. I noticed this issue when I reviewed the first version of the manuscript, but somehow, it failed to make my list of comments...

   *Response:*
   Thanks. Eddy fluxes are downgradient! I have rephrased that line.

2. Lines 70 and 392: "killing" is a strong word. Can something less emotive be used? How about "dissipation"?

   *Response:*
   I think 'killing' is okay in both of these contexts. The paper by Rai is entitled 'Scale of oceanic eddy killing by wind from global satellite observations'.

3. Line 76: I suggest replacing "its use in this" by "the" – much simpler!

   *Response:*
   Done.

4. Equation (6): The second term in the brackets on the right hand side is missing a minus sign.

   *Response:*
   Thanks. Corrected.

5. Line 141: I do not understand how there can be gains due to negative work due to R that will be cancelled out by positive work. This is because, from Figure 2b, the wind work difference is always negative? I would delete the text in this sentence that begins with "whereby".

   *Response:*
   Done.

6. Are the figures shown in Figure 2 computed using the approximation given by (8), or are they exact? One could clarify this in the caption.

> *Response:*
> We use the full expression for this. This is now clarified in the figure caption.

7. Line 144: Surely there are more fundamental (earlier) references that can be given here, notably Duhaut and Straub (2006) and Zhai and Greatbatch (2007)?

> *Response:*
> Have inserted Zhai and Greatbatch (2007).

8. Line 157: I am not comfortable with the word "singular". How about "In this work, we proceed, for simplicity, by representing an eddy using only the first baroclinic mode".

> *Response:*
> Changed.

9. Line 184: I do not like the wording immediately after equation (15). After all, it is not that solutions to (15) can only be found in the limit g'/g -> 0 but rather that the solutions you give are only valid in that limit. I suggest to delete the text that completes the sentence immediately after (15) and, instead, change the sentence that starts on line 185 to. "In the limit g'/g -> 0, the BT is described by etc...".

> *Response:*
> Thanks. Changed.

10. Equation (19): Somewhere you need to say that z is measured positive downward.

> *Response:*
> Done.

11. Line 232: It is not true that gamma governs the stratification of the water column. In fact, it only influences the stratification within the eddy, and it is only a modifiying influence because the basic stratification is set by T_ref.

> *Response:*
> Yep, you are right. We will say 'influencing' instead.

12. Lines 232-233: In general, H_1 is not the point of zero crossing for horizontal velocities. This is only going to be the case in special situations, as is acknowledged on line 244. I do not think any mention of the zero crossing should be given on lines 232-233. Rather, the issue should be discussed in the paragraph beginning on line 243.

> *Response:*
> That makes sense. Text has been appropriately modified.

13. Lines 236-237: I do not see how delta rho can be the density difference between the top and the bottom of the ocean, not least because there is also a contribution from the first term on the right hand side of equation (20).

> *Response:*
> Perhaps this is a bit confusing. Have removed the brackets from N^2(z/g). The first term is a linear profile, and the second term gives the shape of an upper and lower layer profile with a thermocline.

14. Lines 253-254: One could give a reference for the drag coefficient that is used. Is it one of the standard formulae, such as given by Large and Pond?

> *Response:*
> Done.

15. Lines 257-259: I can understand running the model for 10 days to allow the geostrophic adjustment to complete. But what about when the wind is switched on? This will also generate inertial oscillations?

> *Response:*
> Yes, you are right. The wind speed of 7 m/s is moderate so the extent of the inertial oscillations is minimal. The model has been tuned appropriately to deal with this.

16. Equation (26): P_rel, as given by equation (11) is negative. So, I guess P_rel here is actually the modulus of P_rel?

> *Response:*
> Yep. Thanks for pointing that out.

17. Line 339: It would help to refer to equations (11) and (27).

> *Response:*
> Done.

18. Equation (29): It is worth pointing out that in the limit H_1/H_2 -> 0, mu -> g'/g.

> *Response:*
> Have modified text.

19. Equation (31): Strictly speaking the boundary condition is (1/N^2)d phi/dz = 0. This can become important because N^2 typically tends to zero in the surface mixed layer and, potentially also, in the bottom boundary layer.

> *Response:*
> That's a good point. Have added this.

20. Lines 370-371: I assume H here is the local depth and so varies in space?

> *Response:*
> Added 'and so varies in space'.

21. Lines 373-374: Better to write "and locations where the ocean depth is shallower than 300m are not considered".

> *Response:*
> Changed.

22. Line 374: "like" -> "by".

> *Response:*
> Done.

23. Line 384: "slower" -> "lower".

> *Response:*
> Done.

24. Line 393: Better to write "The region between 5 S and 5 N has etc…"

> *Response:*
> Done.

25. Line 408: Is "reductions" correct? Both R_d and |u_a| increase offshore, with opposing effect on the dissipation rate?

> *Response:*
> Thanks. Have said 'most likely due to an increase in R_d and reduction in |u_a|'

26. Line 410: To write "display weaker eddy energy" begs the question "weaker than what"? I guess weaker than in a high resolution ocean-only simulation? However, this needs to be clearly stated. I suggest the text on this topic, including the following line, is revised.

> *Response:*
> Have reworded this bit because I think it may have been confusing.

27. Line 421: "does not vary" -> "are similar" and earlier on this line "pattern" -> "patterns"
    .

> *Response:*
> Changed to 'We also see that the pattern is similar to that seen in Lambda_{rel}(R_d)
> (Fig. 8a,b), particularly across the North West Atlantic and Southern Ocean. This
> is clearly because the spatial pattern of the chosen eddy length scale (R_d or
> (L_e/hat{Rd})R_d) are similar, since (L_e/hat{Rd})R_d depends on R_d.'

28. Line 423: How is hat{lon} defined?

> *Response:*
> Defined.

29. Line 445: Should "winds" be "relative winds"?

> *Response:*
> Changed.

30. Line 486: Better to say that surface modes can be computed over variable topography
    and that the horizontal velocities at the bottom tend to be smaller than if a flat bottom
    is assumed (they are not always zero). My own view on this is that using the standard
    modes, as done by the authors, is more straightforward and is to be preferred.

> *Response:*
> Thanks. Updated text.